# Diagnosing Generalization Failures from Representational Geometry Markers

**Chi-Ning Chou**
Flatiron Institute

**Artem Kirsanov**
Harvard University

**Yao-Yuan Yang**
Google DeepMind

**SueYeon Chung**
Harvard University

## Abstract

Generalization—the ability to perform well beyond the training context—is a hallmark of biological and artificial intelligence, yet anticipating unseen failures remains a central challenge. Conventional approaches often take a "bottom-up" mechanistic route by reverse-engineering interpretable features or circuits to build explanatory models. While insightful, these methods often struggle to provide the high-level, predictive signals for anticipating failure in real-world deployment. Here, we propose using a "top-down" approach to studying generalization failures inspired by medical biomarkers: identifying system-level measurements that serve as robust indicators of a model's future performance. Rather than mapping out detailed internal mechanisms, we systematically design and test network markers to probe structure–function links, identify prognostic indicators, and validate predictions in real-world settings. In image classification, we find that task-relevant geometric properties of in-distribution (ID) object manifolds consistently forecast poor out-of-distribution (OOD) generalization. In particular, reductions in two geometric measures—effective manifold dimensionality and utility—predict weaker OOD performance across diverse architectures, optimizers, and datasets. We apply this finding to transfer learning with ImageNet-pretrained models. We consistently find that the same geometric patterns predict OOD transfer performance more reliably than ID accuracy. This work demonstrates that representational geometry can expose hidden vulnerabilities, offering more robust guidance for model selection and AI interpretability.

## 1 Introduction

Biomarkers—like blood pressure or cholesterol levels—are indispensable tools for anticipating health risks before symptoms emerge. Throughout the history of medicine, physicians have often utilized these diagnostic measures effectively before figuring out all the biological details. [1] This pragmatic, top-down approach of correlating biomarkers with outcomes has thus driven medical progress, while simultaneously providing the foundational insights for figuring out causal mechanisms. In neuroscience, the same methodology has been fruitful: single-neuron and population-level signatures have served as useful analysis units, revealing principles of coding and computation often before a full mechanistic understanding (Rigotti et al., 2013; Barak et al., 2013; Mastrogiuseppe & Ostojic, 2018; Stringer et al., 2019).

As deep neural networks (DNNs) become increasingly integrated into critical applications, a similar challenge arises: how can we anticipate their unseen failures? This is particularly important under distribution shifts where training and deployment environments differ (Sagawa et al., 2020a; Liu et al., 2021; Yang et al., 2024). Current research often favors bottom-up approaches, such as mechanistic interpretability (MI), which aims to reverse-engineer DNNs by identifying interpretable features (Olah et al., 2017; Yun et al., 2023; Cunningham et al., 2023), functional circuits (Olah et al., 2020; Dunefsky et al., 2024), or causal structures (Mueller et al., 2024; Geiger et al., 2025). While

---

Contact: {cchou,schung}@flatironinstitute.org

[1]The lipid hypothesis, for instance, linked cholesterol to cardiovascular disease risk well before lipid pathways were mapped. Similarly, selective serotonin reuptake inhibitors treated depression, informed by the serotonin hypothesis, decades before serotonin's precise role in mood regulation was fully understood.

these MI methods offer granular insights, they may lack identifiability (Méloux et al., 2025), and it remains unclear how they can provide concrete diagnostics on real-world models.

Here we propose a complementary perspective inspired by the history of medicine: a diagnostic, system-level paradigm for understanding neural networks. Rather than attempting to reconstruct their internal mechanisms post-hoc, we focus on developing task-relevant measurements—markers for AI models—that serve as reliable indicators of potential failure modes. Our methodology follows a three-step cycle (Figure 1): (i) **Marker Design**: develop task-relevant measures to probe which structures in neural networks (e.g., feature vectors, weights) relate to their function and performance; (ii) **Prognostic Discovery**[2]: conduct medium-size experiments across diverse architectures and hyperparameters, and identify patterns that serve as prognostic indicators—signals present in in-distribution (ID) properties that can forecast future generalization failures without requiring any knowledge of the out-of-distribution (OOD) tasks; (iii) **Real-world Application**: apply these insights to practical settings, such as predicting which pretrained models will transfer more robustly across datasets. We demonstrate this research cycle by using ID measures based on task-relevant representational geometry to diagnose failure in OOD generalization. Our framework points toward a diagnostic science of AI models, offering tools to anticipate vulnerabilities and improve robustness in safety-critical domains.

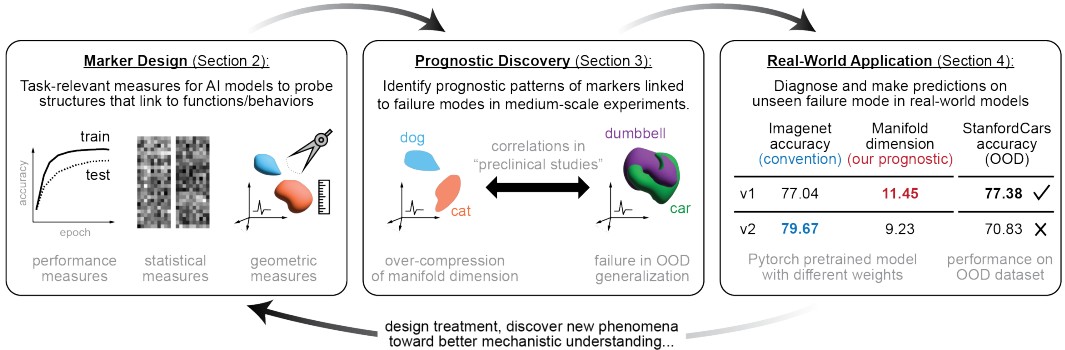

Figure 1: A diagnostic, system-level paradigm for studying generalization failures in DNNs, with an example on image classification. See Section 1.1 for an overview.

## 1.1 OVERVIEW AND OUR CONTRIBUTIONS

In this work, we apply the proposed diagnostic, system-level paradigm to investigate failure modes of OOD generalization in image classification. Our key finding is that feature overspecialization—quantified by reduced effective dimensionality and utility of object manifolds—is a reliable indicator of poor performance under class-level distribution shifts in transfer learning.

**Marker design for image classification (Section 2).** A central step in our diagnostic framework is selecting and designing *markers*—scalar quantities computed entirely from ID data—that capture aspects of a pretrained model relevant for downstream generalization. In image classification, we focus on penultimate-layer feature vectors, as the final classification decision is obtained by a linear readout from this layer. Accordingly, we evaluate a broad family of candidate markers, including: (i) accuracy- and logits-based quantities, (ii) low-order statistical summaries of representations (e.g., sparsity, covariance structure), and (iii) geometric measures of class-conditioned point-cloud manifolds—such as participation ratio, within-class spread, neural-collapse metrics (Papyan et al., 2020; Harun et al., 2025), numerical rank (Masarczyk et al., 2023; Harun et al., 2024), and task-relevant geometric measures from the GLUE framework (Chou et al., 2025a).

**Prognostic discovery of OOD generalization failures (Section 3).** We conducted exploratory experiments to investigate whether these metrics can predict failures in OOD generalization. Specifically, we trained a broad class of deep networks on in-distribution (ID) data (e.g., CIFAR-10) and

---

[2]In medicine, diagnostics identify present conditions, while prognostics forecast future risks. Our framework is termed "diagnostic" broadly, with Step 2 specified as "prognostic discovery" to emphasize prediction of OOD failures from ID data.

evaluated their OOD performance on datasets with disjoint classes (e.g., CIFAR-100; Figure 3a,b). Our sweep spanned five architectures (e.g., ResNet, VGG), multiple depths, two optimization algorithms (SGD, AdamW), and a grid of hyperparameters (learning rate, weight decay). We found that different training hyperparameters can lead to markedly different OOD performance, despite nearly identical ID train and test accuracy. Task-relevant geometric measures of ID object manifolds correlated far more strongly with OOD performance than conventional performance metrics (e.g., ID accuracy) or statistical measures (e.g., sparsity, covariance) (Figure 3c, top). In particular, reductions in effective dimensionality and utility consistently served as prognostic indicators of OOD failure (Figure 3c, bottom). Together with prior work linking representational geometry to feature learning (Chou et al., 2025b), these findings suggest that overspecialized features undermine generalization performance, echoing previous accounts of shortcut learning (Geirhos et al., 2020).

**Applications to failure prediction in pretrained models (Section 4).** Finally, we applied our prognostic indicators to ImageNet-pretrained models from public repositories. In practice, when selecting among multiple pretrained weights of the same architecture, the most common criterion is test accuracy. Here, we measured the effective dimensionality and utility of ImageNet object manifolds from 20 architectures available in PyTorch (e.g., RegNet, MobileNet, WideResNet), each released with two weights (v1 and v2); by construction, v2 achieves higher ID accuracy. Unlike our controlled prognostic studies, these pretrained weights were produced under distinct training recipes, regularization schemes, and preprocessing pipelines, making them a much more heterogeneous testbed. Nevertheless, consistent with the predictions from our medium-scale experiments, models where v1 exhibited higher manifold dimension and utility than v2 also achieved better OOD performance under v1 weights—even though v1 had lower ID accuracy (Figure 5). This demonstrates that ID representational geometry can serve as an early diagnostic for OOD robustness.

*Summary.* Our work demonstrates a diagnostic, system-level paradigm that complements conventional mechanistic interpretability by focusing on predictive indicators of model failure. Our results highlight how task-relevant geometric measures of ID representations can serve as markers for diagnosing failures in OOD generalization, even when mechanistic details remain opaque.

## 1.2 RELATED WORK

**Representational geometry and generalization.** A growing body of work suggests that properties of internal representations in DNNs can indicate generalization performance. In the standard ID setting, both statistical features of activations—such as sparsity, covariance, and inter-feature correlations (Morcos et al., 2018)—and geometric measures of object manifolds (Ansuini et al., 2019; Cohen et al., 2020; Chou et al., 2025b) have been predictive. For example, networks that generalize well often exhibit low intrinsic dimensionality in their final-layer representations, and such compactness correlates with test accuracy in image classification (Ansuini et al., 2019). A related phenomenon is *neural collapse* (Papyan et al., 2020), where within-class variability of final hidden representations vanishes in the terminal phase of training.

The picture becomes more convoluted under distribution shifts. (Galanti et al., 2022) showed that neural collapse can generalize to new data points and classes when trained on sufficiently many classes with lots of samples. By contrast, (Zhu et al., 2023) found that encouraging diversity and decorrelation among features improves OOD performance in image and video classification. Similarly, in neuroscience, high-dimensional yet smooth population codes in mouse visual cortex have been linked to generalization across stimulus conditions (Rigotti et al., 2013; Stringer et al., 2019). These conflicting results call for more systematic study on how representational properties connect to OOD generalization, and our findings—that OOD failures correlate with overcompression of object manifolds—support these results on the advantage of high-dimensional representations. However, most of these approaches rely on generic geometric or statistical descriptors that are not explicitly tied to the computational task, whereas the GLUE measures we employ use the anchor point distribution (Figure 2c and Section B.3) to directly link representational geometry to downstream linear classification performance.

**Distribution shift in ML: detection, transfer, and prior approaches.** OOD detection methods aim to distinguish OOD samples from ID samples by examining differences in their feature representations or logit distributions. These approaches typically operate under label-preserving distribution shifts, where the input distribution changes but the class label space remains the same. In

contrast, class-level OOD generalization—also called transfer learning—is substantially more challenging, since the OOD task contains entirely unseen class labels. Performance in this setting is usually assessed by training a linear probe (often on the penultimate-layer features) of a pretrained network. Several works have studied how architectural or representational factors influence this probe-based OOD performance. For example, the Tunnel Effect papers (Masarczyk et al., 2023; Harun et al., 2024) showed that the drop in OOD linear-probe accuracy across layers correlates with a drop in the numerical rank of OOD features. Similarly, Neural Collapse–based analyses (Harun et al., 2025) have examined how the extent of collapse (e.g., the $\mathcal{NC}1$ metric (Papyan et al., 2020)) relates to OOD generalization performance across layers. Outside image classification, recently Li et al. (2025) used interpretability methods to predict OOD model behavior in language tasks through ID attention patterns.

In this work, we incorporate several of these ideas into our marker design. For OOD detection methods, many algorithms require access to OOD samples in their scoring pipelines, making them incompatible with our ID-only diagnostic setting. However, methods that rely solely on logit statistics or feature-level summaries can be adapted into scalar markers and included in our evaluation. For the Tunnel Effect and Neural Collapse lines of work, we directly implement their corresponding measures—numerical rank and feature-collapse metrics (e.g., $\mathcal{NC}1$)—and compare them against our GLUE-based markers in the prognostic analysis. It is worth noting that both the Tunnel Effect (Masarczyk et al., 2023; Harun et al., 2024) and Neural Collapse (Harun et al., 2025) studies primarily analyze how their measures vary across layers within the same model, rather than across different models or training configurations. Our focus, in contrast, is on comparing models trained under different hyperparameters or initialization regimes, which is the setting relevant for model selection and prognostic prediction.

## 2 MARKERS FOR IMAGE CLASSIFICATION

Given a neural network with parameters $\theta$ and an ID dataset $\mathcal{D}_{\mathsf{ID}}$, we define a marker as a function that maps $(\theta, \mathcal{D}_{\mathsf{ID}})$ to a scalar value indicative of potential failure modes in OOD generalization. Train and test accuracy are examples of such markers, but they are often non-discriminative (D'Amour et al., 2022), propelling us to open the black box of DNNs and seek measures that are both task-relevant and discriminative.

Among the many ways to peer inside a DNN, we focus on feature embeddings. Concretely, we analyze penultimate-layer feature vectors $\{\mathbf{z}_i\}_{i=1}^M$ (e.g., `avgpool` in ResNet, see Table 3) extracted from the ID data. Each $\mathbf{z}_i \in \mathbb{R}^N$ is an $N$-dimensional feature vector, and in image classification these can be grouped by class: letting $P$ denote the number of classes and $M^\mu$ the number of samples in class $\mu$, we write $\{\mathbf{z}_i^\mu\}_{i=1}^{M^\mu}$ so that $\{\mathbf{z}_i\}_{i=1}^M = \bigcup_{\mu=1}^P \{\mathbf{z}_i^\mu\}_{i=1}^{M^\mu}$. In addition to geometric markers derived from representational manifolds, we also consider conventional ID-only markers inspired by prior OOD-detection methods. These include low-order statistical summaries of penultimate representations (e.g., sparsity, covariance structure, pairwise distances and angles) as well as quantities computed directly from the logits distribution (e.g., averaged confidence).

In the remainder of this section, we first review the conventional statistical and logits-based markers that serve as baselines in our analysis (Section 2.1), and then introduce task-relevant geometric markers grounded in representational manifold theory (Section 2.2).

### 2.1 CONVENTIONAL MEASURES

We also examine low-order statistics of penultimate feature vectors. We consider several standard statistics: activation sparsity, off-diagonal covariance magnitude, and mean pairwise distance/angle. Each measure is applied both globally across $\{\mathbf{z}_i\}$ and within each class $\{\mathbf{z}_i^\mu\}$. These descriptors summarize the distribution of representations but do not capture per-class manifold geometry, motivating the measures introduced next. Formal definitions are given in Section B.2.

In addition to feature-level statistics, we incorporate several logit-based markers commonly used in OOD-detection research and adapt them to our ID-only diagnostic setting, including averaged confidence (AUROC) (Hendrycks & Dietterich, 2018), Entropy (Guillory et al., 2021), and Energy (Liu et al., 2020). While these methods were originally designed to detect OOD inputs—and typically

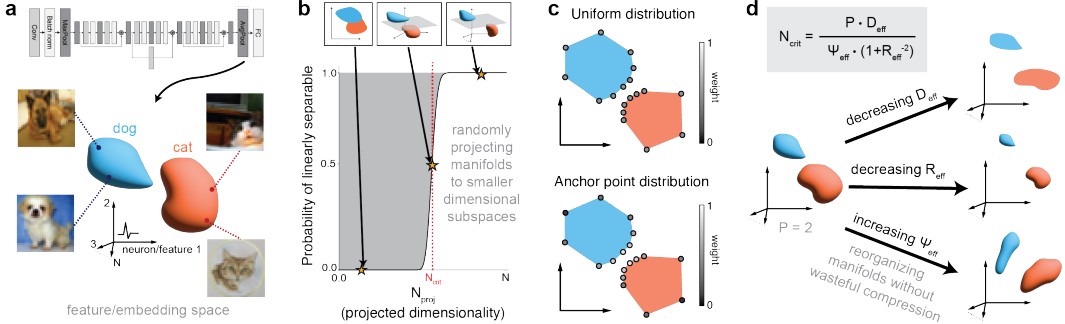

Figure 2: **Object manifolds and task-relevant geometric measures. a,** Object manifolds are the per-class point clouds in the feature space. **b,** Critical dimension $N_{\text{crit}}$ quantifies the degree of manifold untangling/separability in an average-case sense via random projection. **c,** Anchor point distribution gives higher weight to points that are more important for linear classification.[4]**d,** The degree of manifold separation (quantified by critical number of neurons $N_{\text{crit}}$) is analytically linked to three task-relevant geometric measures: effective dimension $D_{\text{eff}}$, radius $R_{\text{eff}}$ and utility $\Psi_{\text{eff}}$.

assume access to shifted data—we evaluate them here as scalar markers derived solely from ID logits.

Beyond these statistical and logits-based metrics, prior work has also analyzed representational geometry in neural populations (Chung & Abbott, 2021; Li et al., 2024). Whereas statistical metrics capture overall spread or pairwise correlations, geometric descriptors characterize manifold structure such as alignment, curvature, and class-specific variability. A widely used task-agnostic geometric marker is the participation ratio (PR), which estimates the intrinsic dimensionality of each class manifold from the spectrum of its covariance matrix. We also include Neural Collapse measures (Papyan et al., 2020; Ammar et al., 2023; Harun et al., 2025) and the numerical rank measure from the Tunnel Effect hypothesis (Masarczyk et al., 2023; Harun et al., 2024). Formal definitions for all markers are provided in Appendix B.

## 2.2 TASK-RELEVANT GEOMETRIC MEASURES

To obtain task-relevant markers of ID representations, we adopt the *Geometry Linked to Untangling Efficiency* (GLUE) framework (Chou et al., 2025a), which builds on the theory of perceptron capacity for points (Gardner & Derrida, 1988) and manifolds (Chung et al., 2018; Wakhloo et al., 2023; Mignacco et al., 2025; Chou et al., 2025a) from statistical physics. Similar to support vector machine (SVM) theory (Cortes & Vapnik, 1995), where the max-margin classifier can be expressed as a linear combination of support vectors, GLUE theory provides an analytic connection between the *critical number of neurons* $N_{\text{crit}}$ and the geometry of object manifolds (Figure 2a) through an *anchor point distribution* over the object manifolds (Figure 2c).

**GLUE as an average-case analog of SVM.** Consider classifying two object manifolds $\mathcal{M}^1 = \text{Hull}(\{\mathbf{z}_1^1, \ldots, \mathbf{z}_M^1\})$ and $\mathcal{M}^2 = \text{Hull}(\{\mathbf{z}_1^2, \ldots, \mathbf{z}_M^2\})$ in $\mathbb{R}^N$, $N_{\text{crit}}$ is defined as the minimum $N_{\text{proj}}$ such that manifolds remain linearly separable with probability at least 0.5 after random projection to an $N_{\text{proj}}$-dimensional subspace (Figure 2b). Manifold capacity $\alpha$ is defined as $P/N_{\text{crit}}$, where $P$ is the number of manifolds. A lower value of $N_{\text{crit}}$ (i.e., higher value of $\alpha$) means that the object manifolds are more separable on average. The key result in GLUE theory is a closed-form formula for $N_{\text{crit}}$:

$$N_{\text{crit}} = \mathop{\mathbb{E}}_{\mathbf{t} \sim \mathcal{N}(0, I_N)} \left[ \max_{\mathbf{s}^1(\mathbf{t}) \in \mathcal{M}^1, \mathbf{s}^2(\mathbf{t}) \in \mathcal{M}^2} \|\text{proj}_{\text{span}(\{\mathbf{s}^1(\mathbf{t}), \mathbf{s}^2(\mathbf{t})\})} \mathbf{t}\|_2^2 \right] \tag{1}$$

where $\mathcal{N}(0, I_N)$ is the isotropic Gaussian distribution in $\mathbb{R}^N$, $\text{span}(\cdot)$ denotes linear span of a set, and $\text{proj}$ denotes orthogonal projection. Equation 1 naturally leads to defining anchor points as the maximizers of the inner optimization problem. The anchor point distribution is a non-uniform

---

[5]This figure is a schematic illustration of the non-uniform, task-relevant anchor point distribution. The 2D depiction is only intuitive and can be misleading, analogous to how in high dimensions Gaussian mass concentrates on the sphere rather than at the origin.

measure over the manifolds and favors those points that are more important for downstream classification (Figure 2c). Hence, GLUE theory can be thought of as an average-case analog of SVM theory: whereas SVM assesses separability in the best-case scenario by leveraging the full feature space, GLUE evaluates separability under random projections, effectively averaging across many such subspaces, and hence is able to capture more complex, heterogeneous, and nuisance structure present in the data (Chou et al., 2025a;b).

By exploiting symmetries in the equation, GLUE theory derives three effective geometric measures—effective dimension $D_{\text{eff}}$, effective radius $R_{\text{eff}}$, and effective utility $\Psi_{\text{eff}}$—and reorganizes Equation 1 into a simple expression (see Section B.3 for details and derivations):

$$N_{\text{crit}} = \frac{P \cdot D_{\text{eff}}}{\Psi_{\text{eff}} \cdot (1 + R_{\text{eff}}^{-2})} \tag{2}$$

where $P$ is the number of manifolds. Intuitively, Equation 2 shows that $N_{\text{crit}}$ decreases (i.e., manifolds become more separable/untangled) with smaller $D_{\text{eff}}$, smaller $R_{\text{eff}}$, and larger $\Psi_{\text{eff}}$ (Figure 2d). Because the GLUE theory captures task-relevant structures in neural representations via the anchor point distribution (as opposed to the uniform distribution, i.e., equiprobable sampling of points), a recent work (Chou et al., 2025b) has shown that $N_{\text{crit}}$ and GLUE measures are much more discriminative than conventional measures (e.g., kernel-based methods, weight changes) in the study of feature learning. GLUE also defines additional measures (e.g., center, axis, center–axis alignment) from the anchor point distribution, detailed in Section B.3 and omitted here for brevity. We provide intuitions for the three effective geometric measures in Table 1 (see Table 4 for the full version).

Table 1: Intuitions for GLUE measures.

|  | $D_{\text{eff}} \geq 0$ | $R_{\text{eff}} \geq 0$ | $\Psi_{\text{eff}} \in [0, 1]$ |
|---|---|---|---|
| Geometric intuition | Quantify the task-relevant dimensionality of object manifolds. | Quantify the task-relevant spread within each manifold relative to their centers. | Quantify the amount of excessive compression of untangling manifolds. |
| Effect on linear separability | More separable when $D_{\text{eff}}$ is small. | More separable when $R_{\text{eff}}$ is small. | More separable when $\Psi_{\text{eff}}$ is large. |
| Example | $D_{\text{eff}}$ equals the dimension of uncor. random spheres | $R_{\text{eff}}$ equals the radius of uncor. random spheres | Collapsing manifolds to points yields $\Psi_{\text{eff}} \to 0$. |
| Interpretation in feature learning[5] | Low $D_{\text{eff}}$ indicates a smaller set of feature modes in use. | Low $R_{\text{eff}}$ indicates more similar feature usage across examples within a class. | Low $\Psi_{\text{eff}}$ indicates inefficient compression of within-class variability. |

**Connection to feature learning.** We follow a top-down view of feature learning (Chou et al., 2025b), where *features* are understood functionally through their consequences for computation (e.g., enabling linear separability) rather than as specific interpretable axes or neurons. This perspective emphasizes how representational geometry changes with feature usage without requiring explicit identification of the features themselves. Moreover, by thinking of a direction in the representation space as a feature (linear representation hypothesis (Park et al., 2024)), the effective geometric measures offer interpretation in feature learning as listed in the table.

## 3   DISCOVER PROGNOSTICS FOR FAILURE IN OOD GENERALIZATION

We study medium-scale models as a testbed for identifying prognostic indicators of failure modes. Our goal is to detect ID signals that reliably predict how a model will behave under distribution shift—without any access to OOD data. This departs from most existing OOD-detection methods, which typically rely on information from the shifted distribution. Our diagnostic analysis uses markers measured solely from ID properties to anticipate vulnerabilities before deployment.

### 3.1   METHODS

We adopt an experimental design in (Chou et al., 2025b) where DNNs are trained on an ID image dataset, and OOD performance is evaluated on a different dataset with a disjoint set of classes.

**Training procedure.** We trained multiple DNN architectures (e.g., ResNet, VGG) from scratch on CIFAR-10. For each architecture, we swept over four initial learning rates, four weight decay

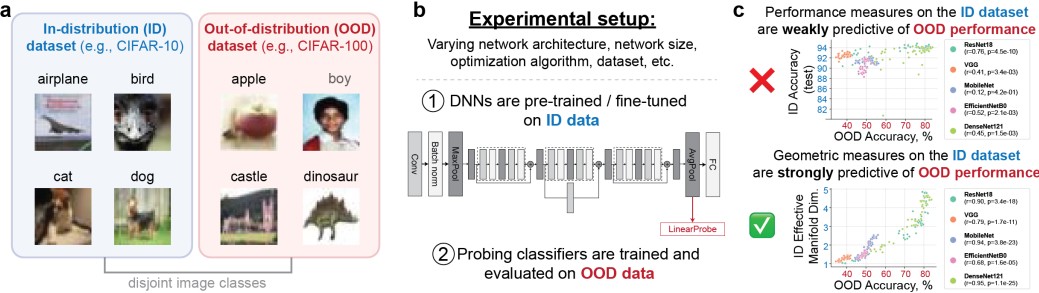

Figure 3: **Prognostic discovery for OOD generalization. a**, We consider the image classification problem with an ID dataset and an OOD dataset with disjoint image classes. **b**, We trained DNNs on the ID dataset and evaluated the OOD performance as linear probe accuracy. **c**, Conventional performance and statistical measures on the ID dataset are weakly predictive of OOD performance, while some task-relevant geometric measures can robustly predict failures in OOD generalization.

values, and three random seeds, using both SGD and AdamW optimizers. In all cases, we ensured that the training accuracy was above 99% and the test accuracy ranged from 88% to 95%.

**OOD evaluation via linear probing.** To assess the OOD generalization of learned representations, we adopt a linear probing framework (Alain & Bengio, 2016; Zhu et al., 2023; Chou et al., 2025b). After ID training, the network's feature extractor was frozen. A new linear classifier was then trained on top of these features using the OOD dataset. The test accuracy of this linear probe served as our measure of OOD performance (Figure 3b). See Appendix A for details.

## 3.2 RESULTS

We find that models trained with distinct hyperparameters can exhibit similar ID accuracy while their OOD performance can differ drastically. This variation, however, is not random; we find that OOD performance can be consistently predicted by geometric properties of ID representations.

**Task-relevant geometric markers are predictive across architectures.** First, we trained different architectures (ResNet, VGG, etc) on CIFAR-10 and evaluated OOD performance on CIFAR-100. As summarized in Figure 4, conventional metrics like ID accuracy and statistical measures like sparsity showed weak and inconsistent correlations with OOD performance. In contrast, several geometric measures—particularly participation ratio, effective dimension, and effective utility—were strong predictors and consistently performed well across all architectures.

**Findings hold across model sizes, optimizers, and datasets.** Next, we tested the generality of our findings by varying model size (ResNet18/34/50), optimizer (SGD, AdamW), and the OOD dataset (CIFAR-100, ImageNet). The results, shown in Figure 4, remained consistent. Across all these settings, task-relevant geometric signatures of the ID data were systematically predictive of OOD performance, whereas alternative markers—including Neural Collapse (Harun et al., 2025), numerical rank (Tunnel Effect (Masarczyk et al., 2023)), and logits-based OOD-detection scores—showed statistically weaker or less consistently predictive trends across settings, with numerical rank performing well in most but a few cases (e.g., VGG-19 using SGD). We suspect this is because the Neural Collapse and Tunnel Effect measures were primarily designed on mathematical intuition rather than task-relevant considerations; as a result, they may not capture the fine-grained structure of complex neural activity patterns across different models or training regimes. Conversely, logit-based markers such as AUROC or entropy are task-relevant but appear to discard too much of the rich information in internal representations, limiting their predictive power in this setting. Additional results are provided in Appendix C.

**Task-relevant geometric markers from ID training data also show strong trends.** While the main figures report results using ID test or validation features, we find that the same geometric indicators measured directly on the ID training data exhibit similarly strong correlations with OOD performance (see Figure 6). This indicates that the predictive signal is not limited to held-out examples, but is already present in the geometry of the training representations themselves.

**ID Test accuracy best predicts OOD performance on corrupted images.** We also consider a corrupted version (e.g., adding noise, varying brightness, pixellating, etc.) of the original images as

an OOD dataset (e.g., CIFAR-10C (Hendrycks & Dieterich, 2018)). Since the class labels remain identical to the ID dataset, OOD performance can be measured directly by the trained network, without training an additional linear probe. In this setting, ID test accuracy is the strongest predictor of performance on corrupted data (see Section C.3), although we note that it does not always work. We also observe distinct geometric patterns across different corruption types. These results highlight that the correlation between OOD accuracy and manifold compression (Figure 4) is non-trivial and specific to class-level shifts, but does not extend to corruption-based shifts where the label space is unchanged. Exploring robustness to corruption thus remains an interesting direction for future work.

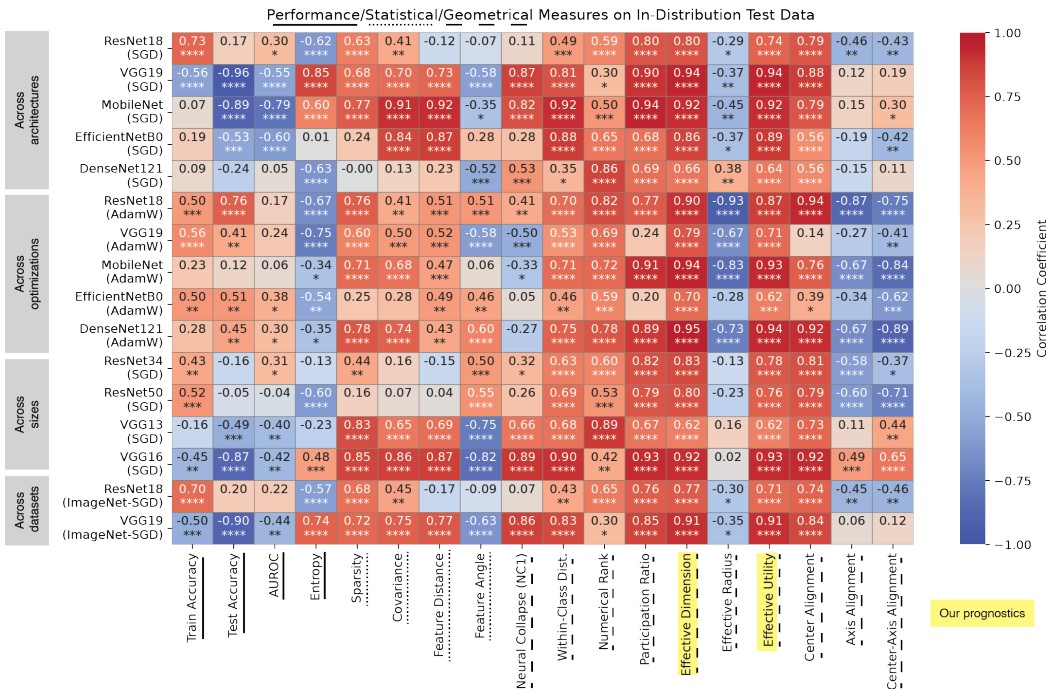

Figure 4: All results on models trained on CIFAR-10, showing correlations between markers (x-axis) and OOD performance across a hyperparameter sweep. Numbers indicate Pearson $r$; asterisks denote significance (* : $p \leq 0.05$; ** : $p \leq 0.01$; *** : $p \leq 0.001$; **** : $p \leq 0.0001$).

## 3.3 DIAGNOSING FAILURES IN GENERALIZATION VIA DETECTING SHORTCUT FEATURES

Failures in generalization are often attributed to a model specializing in its training regime. A classic example is *overfitting*, where high training accuracy but low validation accuracy indicates that the model has memorized the training set rather than learned transferable patterns. Under distribution shift, however, such straightforward indicators as validation accuracy are absent. Our findings in Section 3.2 suggest using $D_{\text{eff}}$ and $\Psi_{\text{eff}}$ measured on ID object manifolds as prognostics for indicating potential failure in OOD datasets with new classes of images.

Failures in OOD generalization are often attributed to reliance on *shortcut* or *spurious* features (Geirhos et al., 2020; Sagawa et al., 2020b; Singla & Feizi, 2021; Yang et al., 2022). A network may correctly classify cows in typical training images, yet fail when cows appear in unusual contexts such as beaches or mountains, suggesting that background cues like grass had been used as unintended predictors of class identity (Beery et al., 2018). Features such as "grass" correspond to microscopic details, whereas generalization performance is a macroscopic outcome. Effective geometric measures act as *mesoscopic descriptors*, bridging how microscopic features are at play and how efficiently they are used for macroscopic behavior, such as separability (Chou et al., 2025b) (see also Table 4). Low $D_{\text{eff}}$ and $\Psi_{\text{eff}}$ indicate that the model relies on a smaller set of features, used inefficiently for separability, agreeing with shortcut-learning interpretations.

Finally, we remark that although untrained or randomly initialized networks also exhibit very high manifold dimension and poor generalization (Chou et al., 2025b) (i.e., lazy learning, Figure 7), our

analysis concerns models with comparable ID validation accuracy—i.e., after meaningful feature learning has taken place. In this regime, larger $D_{\text{eff}}$ and $\Psi_{\text{eff}}$ reflect richer task-relevant variability, whereas excessive compression signals overspecialization to the ID distribution.

# 4 APPLICATIONS TO PREDICTING PERFORMANCE OF TRANSFER LEARNING

A common scenario in applied machine learning involves selecting a pretrained model from a public repository like PyTorch Hub or Hugging Face. For a given architecture, multiple sets of weights are often available, each trained with different optimization recipes, regularization schemes, or data preprocessing pipelines. The standard heuristic is to choose the model with the highest reported in-distribution (ID) accuracy. However, it is unclear whether this metric reliably predicts performance on other downstream tasks, especially under the distribution shifts inherent in transfer learning.

Here, we apply the prognostic indicators discovered in our exploratory experiments (Section 3) to this practical challenge. Our findings suggest a clear guiding principle for model selection: when faced with multiple weights for the same architecture, **prefer the model that exhibits higher effective manifold dimensionality ($D_{\text{eff}}$) and utility ($\Psi_{\text{eff}}$) on its ID data**, as this signals a greater potential for robust OOD generalization.

| Model | | Marker | | ID Acc | OOD Acc (measured by a linear probe) | | | | | | | | | OOD pred. acc |
| --- | --- | --- | --- | --- | --- | --- | --- | --- | --- | --- | --- | --- | --- | --- |
| Arch. | Weight | $D_{\text{eff}}$ | $\Psi_{\text{eff}}$ | ImageNet | Pet | Flower | Food | Nat. | Place | Car | Texture | Aircraft | Skin | |
| MobileNet (V3_Large) | v1 | **13.07** | **0.956** | 74.04 | 89.16 | **89.71** | **72.79** | 36.56 | **44.39** | **53.28** | **67.64** | **41.60** | **80.29** | 7/9 |
| | v2 | 10.06 | 0.924 | **75.27** | **91.05** | 85.82 | 71.94 | **37.66** | 43.70 | 49.97 | 66.28 | 34.86 | 80.12 | |
| RegNet_X (16GF) | v1 | **10.50** | **0.929** | 80.06 | 92.72 | **88.74** | **75.86** | **42.09** | 45.72 | **56.13** | 70.96 | **41.77** | **82.65** | 6/9 |
| | v2 | 8.78 | 0.899 | **82.72** | **93.13** | 85.88 | 74.95 | 38.84 | **46.02** | 49.43 | **72.30** | 28.01 | 80.49 | |
| RegNet_X (1_6GF) | v1 | **11.35** | **0.935** | 77.04 | 91.89 | **88.28** | **72.36** | **39.33** | **44.27** | **53.96** | **68.60** | **40.37** | **79.45** | 8/9 |
| | v2 | 9.19 | 0.908 | **79.67** | **92.43** | 79.22 | 68.03 | 32.68 | 43.48 | 40.46 | 67.06 | 22.99 | 78.26 | |
| RegNet_X (32GF) | v1 | **10.15** | **0.930** | 80.62 | 93.00 | **88.06** | **76.00** | **40.93** | 45.72 | **54.86** | **71.67** | **38.54** | **82.68** | 7/9 |
| | v2 | 8.81 | 0.905 | **83.01** | **93.38** | 85.94 | 75.73 | 38.59 | **46.35** | 47.22 | 70.41 | 26.42 | 81.19 | |
| RegNet_Y (16GF) | v1 | **9.87** | **0.923** | 80.42 | **92.81** | **88.19** | **76.66** | **41.08** | 45.89 | **54.62** | 71.10 | **41.60** | **83.42** | 7/9 |
| | v2 | 9.29 | 0.906 | **82.89** | 92.30 | 85.07 | 75.99 | 37.58 | **46.72** | 48.63 | **71.19** | 27.26 | 80.02 | |
| RegNet_Y (1_6GF) | v1 | **11.06** | **0.933** | 77.95 | 92.30 | **87.46** | **73.64** | **40.26** | **44.72** | **53.89** | **69.17** | **42.75** | **80.49** | 8/9 |
| | v2 | 8.60 | 0.909 | **80.88** | **93.13** | 79.46 | 69.61 | 31.87 | 43.67 | 40.58 | 65.11 | 25.86 | 77.56 | |
| RegNet_Y (400MF) | v1 | **10.98** | **0.933** | 74.05 | 90.55 | **84.88** | **68.07** | **34.74** | **42.68** | **43.00** | **65.60** | **38.32** | **76.56** | 8/9 |
| | v2 | 9.16 | 0.914 | **75.80** | **91.06** | 76.81 | 63.61 | 27.75 | 40.77 | 33.52 | 63.71 | 22.36 | 75.92 | |
| RegNet_Y (800MF) | v1 | **11.27** | **0.937** | 76.42 | 91.79 | **86.25** | **72.28** | **38.97** | **44.34** | **52.91** | **69.34** | **42.29** | **79.09** | 8/9 |
| | v2 | 9.42 | 0.913 | **78.83** | **92.44** | 79.48 | 68.00 | 32.28 | 42.92 | 43.37 | 65.80 | 30.10 | 77.39 | |
| RegNet_Y (8GF) | v1 | **10.32** | **0.926** | 79.34 | 92.40 | **88.23** | **76.39** | **42.38** | 45.86 | **55.70** | 70.09 | **44.32** | **83.65** | 6/9 |
| | v2 | 8.61 | 0.903 | **81.68** | **93.15** | 85.10 | 75.53 | 38.19 | **46.24** | 50.16 | **70.60** | 28.30 | 80.99 | |
| ResNet (152) | v1 | **9.48** | **0.920** | 78.31 | **92.86** | **84.73** | **71.88** | **36.26** | 44.81 | **54.65** | 67.73 | **35.47** | **81.22** | 7/9 |
| | v2 | 9.14 | 0.910 | **82.28** | 92.66 | 81.17 | 71.70 | 32.23 | **46.02** | 40.62 | **69.82** | 23.16 | 79.19 | |
| ViT (B_16) | v1 | **9.56** | **0.956** | 81.87 | **95.24** | **99.56** | **94.04** | **61.09** | **50.51** | **79.98** | **78.51** | 43.05 | 81.42 | 7/9 |
| | v2 | 8.86 | 0.928 | **85.30** | 95.18 | 97.71 | 89.98 | 60.03 | 50.14 | 77.93 | 76.12 | **50.10** | **83.58** | |
| ViT (H_14) | v1 | **9.89** | **0.946** | 85.71 | 96.33 | **99.64** | **95.95** | 59.60 | 52.02 | 85.48 | 77.66 | 46.91 | 83.98 | 1/9 |
| | v2 | 8.32 | 0.907 | **88.55** | **96.73** | **99.64** | 95.60 | **69.76** | **53.29** | **88.88** | **79.73** | **55.17** | **84.75** | |
| ViT (L_16) | v1 | **9.56** | **0.947** | 85.15 | 95.99 | **99.58** | **95.74** | 58.39 | 51.37 | 84.16 | **79.15** | 45.48 | **81.78** | 4/9 |
| | v2 | 6.49 | 0.888 | **88.06** | **96.51** | 99.43 | 95.02 | **66.18** | **51.95** | **87.20** | 78.35 | **55.84** | 81.45 | |
| WideResNet (50_2) | v1 | 9.19 | 0.914 | 78.47 | 92.02 | 81.14 | 67.46 | 32.46 | 43.26 | 42.34 | 66.81 | 27.64 | **79.92** | 8/9 |
| | v2 | **10.59** | **0.925** | **81.60** | **93.20** | **82.29** | **72.67** | **35.41** | **46.61** | **47.50** | **70.90** | **28.03** | 79.89 | |

Figure 5: Predict OOD transfer performance on ImageNet-pretrained models via $D_{\text{eff}}$ and $\Psi_{\text{eff}}$. For the first block of models, our prognostic indicators predicted that v1 would outperform v2. For the second block of models, our prognostic indicators predicted the other way around.

**Experimental procedure.** To test this principle, we analyzed 20 popular architectures from PyTorch's official repository, each released with two sets of weights (v1 and v2). By design, the v2 weights achieve higher accuracy on the ID ImageNet benchmark. However, the specific changes in training procedure are often opaque to the end-user (see Table 5 for key differences). This heterogeneity makes for a challenging and realistic testbed for our diagnostic framework. For each v1/v2 pair, we first measured the $D_{\text{eff}}$ and $\Psi_{\text{eff}}$ of their ImageNet object manifolds. We then evaluated their OOD transfer performance on 9 image classification datasets: Flowers102 (Nilsback & Zisserman, 2008), Stanford Cars (Krause et al., 2013), Places365 (Zhou et al., 2017), Food101 (Bossard et al., 2014), Oxford-IIIT Pet (Parkhi et al., 2012), etc. For each OOD dataset, we train a linear probe on the training set of the OOD dataset, and report the test accuracy (see Section 3.1 for details). See Appendix D for more experimental details.

**Diagnosing transferability through ID effective manifold geometry.** Consistent with the hypothesis derived from our initial explorations, we found that models with higher $D_{\text{eff}}$ and $\Psi_{\text{eff}}$ of-

ten demonstrated stronger OOD transfer performance, even when their ID ImageNet accuracy was lower. As shown in Figure 5, across the 20 architectures we examined, our prognostic indicators predicted that v1 would outperform v2 on OOD transfer in 14 cases (despite v2 having higher ID accuracy), that v2 would outperform v1 in 1 case, and yielded no clear verdict for the remainder. Among these 15 models and 9 OOD datasets, our prediction accuracy is 73.02% (92 out of 126). This is much higher than using ID test accuracy as a predictor for OOD performance (37.22%). We remark that using some of the other markers (e.g., Neural Collapse, Participation Ratio) also yields non-trivial prediction accuracy in OOD performance. See Section D.4 for more results.

**Revealing differences in fine-tuning dynamics.** Finally, we explored whether these initial feature advantages persist during full-model fine-tuning. As expected from prior work showing that the benefits of pretraining diminish with longer fine-tuning (Kornblith et al., 2019; He et al., 2018), both v1 and v2 initializations ultimately converged to a similar performance level. However, we observed a drastic difference in the early fine-tuning stages: models initialized with v1 weights sometimes exhibited faster learning, hinting that their features may provide a more efficient transferable starting point (Figures 22, 23). These results show that test-relevant geometric measures can reveal differences in fine-tuning dynamics, motivating future study on their role in transfer learning.

## 5    DISCUSSION

We introduced a diagnostic, system-level paradigm for anticipating generalization failure in neural networks. Instead of reconstructing detailed internal mechanisms, we treated task-relevant geometric markers of ID representations as prognostic indicators. Through discovering prognostic markers in medium-sized experiments, we found that overcompression of object manifold dimension consistently predicts failures in OOD generalization. Applied to ImageNet-pretrained models—a far more heterogeneous real-world setting—our prognostic measures predict which models transfer more robustly across tasks. Together, these results demonstrate the power of a diagnostic framework for studying generalization failures. This work opens up several future directions.

- **Theoretical foundations.** In Section 3.3, we link overcompression of object manifolds to overspecialization of learned features. Strengthening the theoretical basis of this hypothesis is important. A related question is whether incorrectly classified OOD examples share common traits that can be explained by the overspecialization intuition.

- **Causal mechanisms and interventions.** Geometric indicators could inspire investigation into underlying causal mechanisms and practical interventions, such as geometry-aware regularization, early-stopping criteria, or model selection rules that prioritize robustness alongside accuracy.

- **Extending the proposed diagnostic research framework.** Expanding our proposed analysis framework beyond vision to language, reinforcement learning, or multi-modal models remains an open challenge. A natural starting point is to first identify and characterize the relevant failure modes in each domain, and then examine how representational markers correlate with those failures. Another direction is to extend our findings into deployable protocols for diagnosing OOD failures across a wider range of models and datasets.

- **Linking diagnostics to parameter transfer.** A future direction is to explore whether insights from our controlled experiments can inform parameter transfer between models of different scales, as in Net2Net (Chen et al., 2015). While our focus here is on diagnostics, connecting to weight transfer could provide a complementary path for robust initialization.

- **Parallels with neuroscience.** High-dimensional yet structured codes in the brain have been linked to generalization in neuroscience studies. Our hypothesis linking manifold compression to feature overspecialization may provide a framework for interpreting these findings and exploring common principles across biological and artificial systems.

Theoretical work on neural networks has long been shaped by mathematics and physics, with an emphasis on bottom-up mechanistic explanations. We suggest that the history of medicine offers a complementary perspective: effective diagnostics can anticipate risks and guide treatment well before underlying causal mechanisms are fully understood. Neural networks, as emergent high-dimensional systems, may likewise benefit from a diagnostic science that anticipates vulnerabilities and guides future mechanistic insight.

ACKNOWLEDGMENTS

We thank Hang Le for the helpful discussion. This work was supported by the Center for Computational Neuroscience at the Flatiron Institute, Simons Foundation. S.C. was partially supported by a Sloan Research Fellowship, a Klingenstein-Simons Award, and the Samsung Advanced Institute of Technology project, "Next Generation Deep Learning: From Pattern Recognition to AI." All experiments were performed using the Flatiron Institute's high-performance computing cluster. Yao-Yuan Yang worked in an advisory capacity.

CODE AVAILABILITY

All code required to reproduce the figures presented is available under an MIT License at https://github.com/chung-neuroai-lab/ood-generalization-geometry

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

## A    EXPERIMENTAL SETTINGS

In this section, we provide a complete description of our experimental setup to facilitate reproducibility.

### A.1    DATASETS

Our study utilized a range of standard image classification datasets, which served different roles: either as in-distribution (ID) training sources or out-of-distribution (OOD) evaluation benchmarks across two distinct experimental settings. Table 2 provides a summary of these roles. Below, we describe each dataset and the specific preprocessing pipelines applied.

Table 2: Summary of dataset roles in our experiments.

| Experimental Setting | In-Distribution (ID) Dataset | Out-of-Distribution (OOD) Datasets |
|---|---|---|
| **Prognostic discovery** (Section 3) | CIFAR-10 | CIFAR-100 ImageNet-1k (resized to 32x32) |
| **Transfer Learning Applications** (Section 4) | ImageNet-1k | Flowers102 Stanford Cars Places365 Oxford-IIIT Pets Food-101 iNaturalist 2018 DTD FGVC-Aircraft HAM10000 |

**Datasets for prognostic discovery.**    In our controlled medium-scale experiments, we trained models from scratch on a single ID dataset and evaluated their generalization to two different OOD datasets with disjoint classes.

- **CIFAR-10** (Krizhevsky et al., 2009) served as our primary **in-distribution (ID)** dataset for training. It contains 60,000 color images of $32 \times 32$ pixels, split into 50,000 training and 10,000 test images across 10 object categories. For training, we normalized images using a per-channel mean of $(0.4914, 0.4822, 0.4465)$ and a standard deviation of $(0.2023, 0.1994, 0.2010)$. We also applied standard data augmentation: padding with 4 pixels on each side, followed by a random $32 \times 32$ crop and a random horizontal flip with 50% probability. For evaluating ID test accuracy, augmentation was disabled.

- **CIFAR-100** (Krizhevsky et al., 2009) was used as the primary **out-of-distribution (OOD)** benchmark. It has the same image format and size as CIFAR-10 but contains 100 distinct object classes with no overlap. For OOD evaluation, images were only normalized using the CIFAR-10 statistics; no data augmentation was applied to ensure a deterministic evaluation protocol.

- **ImageNet-1k** (Deng et al., 2009) was used as a second, more challenging **OOD** benchmark to test generalization under a significant domain shift. This dataset contains over 1.2 million high-resolution images from 1,000 categories. To maintain compatibility with our CIFAR-trained models, all ImageNet images were resized to $32 \times 32$ pixels using bicubic interpolation. They were then normalized using the standard ImageNet per-channel mean $(0.485, 0.456, 0.406)$ and standard deviation $(0.229, 0.224, 0.225)$. No data augmentation was applied during evaluation.

**Datasets for Transfer Learning Applications.**    In this setting, we analyzed publicly available models pretrained on ImageNet-1k and evaluated their transferability to three downstream, fine-grained classification tasks.

- **ImageNet-1k** served as the **in-distribution (ID)** dataset, as all models we analyzed were pretrained on it. For measuring the ID geometric markers, we used the official validation set. Images

were processed according to the standard pipeline for each model: resized to $256 \times 256$, center-cropped to $224 \times 224$, and normalized using the standard ImageNet mean and standard deviation.

- **Flowers102** (Nilsback & Zisserman, 2008) is a fine-grained OOD dataset containing 8,189 images of flowers belonging to 102 different categories.
- **Stanford Cars** (Krause et al., 2013) is another fine-grained OOD dataset consisting of 16,185 images of cars, categorized by 196 classes (e.g., make, model, and year).
- **Places365** (Zhou et al., 2017) is a large-scale scene-centric OOD dataset with over 1.8 million images from 365 scene categories.
- **Oxford-IIIT Pets** (Parkhi et al., 2012) contains a 37-category pet dataset with roughly 200 images for each class.
- **Food-101** (Bossard et al., 2014) includes 101,000 images of 101 food dishes (750 training and 250 test images per class). The dataset exhibits large variation in presentation, lighting, and style.
- **iNaturalist 2018** (Van Horn et al., 2018) consists of over 450,000 training images from more than 8,000 species of plants, animals, and fungi, collected and verified by citizen scientists on the iNaturalist platform. The long-tailed distribution and diverse real-world conditions make this dataset highly challenging for transfer evaluation.
- **Describable Textures Dataset (DTD)** (Cimpoi et al., 2014) contains 5,640 texture images annotated with 47 describable texture attributes. Images span varied materials, lighting, and scales.
- **FGVC-Aircraft** (Maji et al., 2013) is a fine-grained visual classification dataset containing 10,000 images across 100 aircraft variants. Images differ in viewpoint, environment, and model-year variations.
- **HAM10000** (Tschandl et al., 2018) is a dermatology image dataset containing 10,015 dermatoscopic images drawn from seven diagnostic categories (e.g., melanocytic nevi, melanoma, benign keratosis, vascular lesions). The images exhibit substantial variation in acquisition conditions, anatomical location, and lesion appearance, making HAM a visually and semantically distinct OOD dataset relative to natural-image pretraining.

For all three OOD datasets in this setting, images were resized to $224 \times 224$ pixels using bicubic interpolation and then normalized. During the OOD evaluation via linear probing, no data augmentation was applied. For the full-model fine-tuning experiments (see Figure 22), data augmentation was applied during the training phase, which included random horizontal flipping (with a 50% probability) and color jitter. These augmentations were disabled during the evaluation of model checkpoints on the OOD validation subsets.

## A.2 MODEL ARCHITECTURES

### A.2.1 MODELS FOR PROGNOSTIC DISCOVERY (SECTION 3)

To ensure our findings generalize across different model design philosophies, our exploratory studies included a diverse set of convolutional neural network (CNN) architectures. All models were adapted for CIFAR-scale ($32 \times 32$ pixel) inputs and trained from random initialization, ensuring that their learned representations were not influenced by prior pretraining:

- **ResNet** (He et al., 2016): A family of foundational deep residual networks that utilize skip connections to enable effective training of very deep models. We used the ResNet-18, ResNet-34, and ResNet-50 variants.
- **VGG** (Simonyan & Zisserman, 2015): Classic deep feedforward networks characterized by their architectural simplicity and sequential stacking of small $3 \times 3$ convolutions. We included VGG-13, VGG-16, and VGG-19, each augmented with batch normalization.
- **MobileNetV1** (Howard et al., 2017): A lightweight architecture designed for computational efficiency through the use of depthwise separable convolutions.
- **EfficientNet-B0** (Tan & Le, 2019): A modern, highly efficient model that systematically scales network depth, width, and resolution using a compound scaling method.
- **DenseNet** (Huang et al., 2017): An architecture designed to maximize feature reuse and improve gradient flow by connecting each layer to every other subsequent layer within dense blocks.

This selection spans a wide architectural landscape, including canonical residual and feedforward designs, modern efficient networks, and architectures with alternative connectivity patterns. This diversity allows us to validate that our findings are a general property of deep representations, rather than an artifact of a specific model family.

### A.2.2 MODELS FOR TRANSFER LEARNING APPLICATIONS (SECTION 4)

For the experiments in Section 4, we shifted from training smaller-scale models from scratch across a wide range of hyperparameters to analyzing publicly available, pretrained models to test our diagnostic framework in a realistic setting. Our primary selection criterion was the availability of two official pretrained weight versions, typically labeled "v1" and "v2", within the PyTorch model repository.

This v1/v2 setup provides a unique opportunity for a controlled comparison. By design, the v2 weights offer higher in-distribution (ID) accuracy on ImageNet, often due to improved training recipes, data augmentation (e.g., AutoAugment), or regularization (e.g., label smoothing). This allows us to directly test our central hypothesis: whether ID geometric markers can identify cases where higher ID accuracy masks a hidden vulnerability, leading to poorer out-of-distribution (OOD) transferability.

Our final set of 20 architectures is highly diverse, spanning multiple design generations and principles. In addition to deeper variants of models used in our control studies (**ResNet-50/101/152**, **MobileNetV2/V3**, **EfficientNet-B1**), our selection also includes:

- **RegNet** (Radosavovic et al., 2020): A family of networks (e.g., RegNetY-400MF, RegNetX-32GF) whose structure is discovered by optimizing a data-driven design space, resulting in well-performing models.
- **ResNeXt** (Xie et al., 2017): An evolution of ResNet that introduces a cardinality dimension, increasing model capacity by aggregating a set of parallel transformations.
- **Wide ResNet** (Zagoruyko & Komodakis, 2016): A variant of ResNet that is wider but shallower, demonstrating that width can be a more effective dimension for improving performance than depth.

### A.3 COMPUTING RESOURCES

All experiments were conducted on NVIDIA H100 (80GB) or A100 (80GB) GPUs, paired with a 128-core Rome CPU and 1 TB of RAM. Training each model for 200 epochs required approximately 1–3 hours, depending on the architecture and optimizer. Unless otherwise specified, all experiments were run on a single GPU worker. These specifications, together with the full training configurations described in earlier subsections, are provided to facilitate reproducibility.

# B  DETAILS ON ID MEASURES

In this section, we define the performance, statistical, and geometric measures used in our analysis. These are computed on the feature representations extracted from models using the ID training dataset, unless stated otherwise. Our goal is to identify which properties of a model's ID representations can serve as reliable indicators of its out-of-distribution (OOD) generalization capability.

The measures are grouped into three categories: **performance** measures that quantify classification accuracy, **statistical** measures that summarize low-order distributional properties of features, and **geometric** measures that characterize the structure of class-specific feature manifolds. A key distinction is that while statistical metrics typically operate on pooled features, our primary geometric measures are computed on object manifolds – the per-class point clouds in representation space. This allows them to directly capture properties relevant to classification, such as manifold size, shape, and correlation structure in the representational space.

We first describe how feature representations are extracted and then define each measure in detail.

## B.1  REPRESENTATION EXTRACTION

All representational measures are computed on feature vectors extracted from the penultimate layer of each network – the final layer before the classification head. This layer captures high-level, task-specialized features that are not yet collapsed into class logits. For convolutional networks, the feature vector is obtained via global average pooling. The exact layers used for each architecture are listed in Table 3.

Table 3: Exact layer names used for extracting feature representations.

| Architecture | Layer name in PyTorch module |
|---|---|
| VGG13 | `features.34` |
| VGG16 | `features.43` |
| VGG19 | `features.52` |
| ResNet | `avgpool` |
| DenseNet121 | `avg_pool2d` |
| MobileNet | `avg_pool2d` |
| EfficientNetB0 | `adaptive_avg_pool2d` |
| RegNet | `avgpool` |
| ResNeXt | `avgpool` |
| Wide ResNet | `avgpool` |

Given an ID dataset $\mathcal{D}_{\mathrm{ID}}$ and a trained network $f_\theta$, let $\mathbf{z}_i \in \mathbb{R}^N$ denote the $N$-dimensional feature vector for the $i$-th input sample $\mathbf{x}_i$ in $\mathcal{D}_{\mathrm{ID}}$, extracted from the layer listed in Table 3. All statistical and geometric measures described in the following subsections are computed from the collection $\{\mathbf{z}_i\}_{i=1}^M$ of such feature vectors, where $M$ is the total number of samples in $\mathcal{D}_{\mathrm{ID}}$.

For measures that require class-specific statistics (e.g., within-class covariance, manifold radius), we further partition $\{\mathbf{z}_i\}$ by ground-truth label into $\{\mathbf{z}_i^\mu\}_{i=1}^{M^\mu}$ for each class $\mu \in \{1, \ldots, P\}$, where $M^\mu$ is the number of samples in class $\mu$.

## B.2  STATISTICAL METRICS

We compute a set of statistical descriptors from the ID feature representations to quantify basic structural properties of the learned embedding space. All metrics are computed from the collection of penultimate-layer feature vectors $\{\mathbf{z}_i\}_{i=1}^M$ extracted from the ID dataset (see Table 3).

**Activation sparsity.**  The activation sparsity measures the proportion of non-zero entries across all feature vectors,

$$\text{sparsity} = \frac{1}{MN} \sum_{i=1}^M \sum_{j=1}^N \mathbf{1}(|z_{ij}| > \varepsilon),$$

where $N$ is the feature dimension and $\varepsilon = 10^{-6}$ is a small threshold to account for numerical noise. Higher sparsity indicates more silent units on average across the dataset.

**Covariance magnitude.** We compute the empirical covariance matrix $\Sigma \in \mathbb{R}^{N \times N}$ over features and take the mean absolute value of its off-diagonal entries,

$$\text{mean\_covariance} = \frac{2}{N(N-1)} \sum_{j < k} |\Sigma_{jk}|,$$

which reflects the average degree of linear correlation between distinct feature dimensions.

**Pairwise distance.** We compute the mean Euclidean distance between all pairs of feature vectors,

$$\text{mean\_distance} = \frac{2}{M(M-1)} \sum_{i < j} \|\mathbf{z}_i - \mathbf{z}_j\|_2,$$

providing a coarse measure of spread in the representation space.

**Pairwise angle.** After $\ell_2$-normalizing each feature vector, we compute cosine similarities and convert them to angles in radians via $\theta_{ij} = \arccos(\cos\_sim_{ij})$. The mean pairwise angle reflects the typical directional separation between features.

All statistical metrics are computed on the raw feature vectors without centering unless required by the measure (e.g., covariance).

### B.3 Geometric measures: participation ratio and GLUE-based task-relevant metrics

Unlike the statistical measures described above, our geometric analysis operates on *object manifolds*—point clouds in feature space containing activations from the same class. This distinction is important: geometric metrics explicitly quantify per-class representational structure, whereas most statistical metrics aggregate across the entire dataset without regard to class boundaries.

**Participation ratio (PR).** As a conventional baseline for manifold dimensionality, we compute the *participation ratio* (PR) of the penultimate-layer features for each class. Let $\{\mathbf{z}_i^\mu\}_{i=1}^{M^\mu}$ denote the $M^\mu$ feature vectors for the $\mu$-th class, and $\lambda_i^\mu$ be the eigenvalues of their covariance matrix. The PR of this class is defined as

$$D_{\mathsf{PR}}^\mu = \frac{(\sum_i \lambda_i^\mu)^2}{\sum_i (\lambda_i^\mu)^2}, \tag{3}$$

which measures the effective number of principal components with substantial variance. In all figures we present the average of PR over all classes, i.e., $\frac{1}{P} \sum M^\mu D_{\mathsf{PR}}^\mu$. While PR is widely used, it is *task-agnostic* and does not incorporate information about class separability.

**Neural Collapse measure (NC1).** In addition to per-class geometric descriptors, we also include a global Neural Collapse–inspired measure that captures the degree of *zero-collapse* between within-class and between-class structure (Papyan et al., 2020; Harun et al., 2025). Let $\Sigma_W \in \mathbb{R}^{N \times N}$ denote the pooled within-class covariance and $\Sigma_B \in \mathbb{R}^{N \times N}$ the between-class covariance of the penultimate-layer features (see Section B.2 for definitions), and let $P$ be the number of classes. We first form a truncated pseudo-inverse of $\Sigma_B$ by eigendecomposition. Write

$$\Sigma_B = U\Lambda U^\top, \quad \Lambda = \text{diag}(\lambda_1, \ldots, \lambda_N), \quad \lambda_1 \geq \cdots \geq \lambda_N \geq 0,$$

and let $\lambda_{\max} = \lambda_1$. We retain only eigen-directions with sufficiently large eigenvalues,

$$\mathcal{I} = \{\, i \,:\, \lambda_i \geq \tau \lambda_{\max} \,\},$$

with a small threshold $\tau$ (we use $\tau = 10^{-3}$ in all experiments), and define the truncated pseudo-inverse

$$\Sigma_B^\dagger = \sum_{i \in \mathcal{I}} \lambda_i^{-1} \mathbf{u}_i \mathbf{u}_i^\top,$$

where $\mathbf{u}_i$ denotes the $i$-th column of $U$. The NC1 (zero-collapse) score is then

$$\mathrm{NC1} = \frac{1}{P}\operatorname{tr}(\Sigma_W \Sigma_B^\dagger).$$

Smaller values of NC1 indicate stronger collapse of within-class variability relative to the between-class structure. We treat NC1 as a geometric marker and compare it with the participation ratio, numerical rank, and the GLUE-based task-relevant measures in our prognostic analysis.

**Tunnel Effect: numerical rank.** Inspired by recent studies on the Tunnel Effect hypothesis (Masarczyk et al., 2023; Harun et al., 2024), we also compute the *numerical rank* of the feature representations. For a given class $\mu$, let $\{\mathbf{z}_i^\mu\}_{i=1}^{M^\mu}$ denote its feature vectors and let

$$\Sigma_\mu = \frac{1}{M^\mu}\sum_{i=1}^{M^\mu}(\mathbf{z}_i^\mu - \mathbf{c}_\mu)(\mathbf{z}_i^\mu - \mathbf{c}_\mu)^\top$$

be the corresponding empirical covariance matrix, where $\mathbf{c}_\mu$ is the class-mean representation (defined above). Let $\sigma_1^\mu \geq \sigma_2^\mu \geq \cdots$ denote the singular values of $\Sigma_\mu$. Following prior work, the numerical rank of class $\mu$ is defined as

$$\mathrm{Rank}_{\mathrm{num}}^\mu = \#\left\{i : \sigma_i^\mu \geq \tau\,\sigma_1^\mu\right\}, \quad \text{with } \tau = 10^{-3}.$$

The reported value is the average over all classes, $\mathrm{Rank}_{\mathrm{num}} = \frac{1}{P}\sum_{\mu=1}^{P}\mathrm{Rank}_{\mathrm{num}}^\mu$. Lower numerical rank indicates stronger compression of the class manifold. Prior work has shown that layers exhibiting low rank often display degraded OOD linear-probe accuracy. We include numerical rank as a baseline geometric marker for comparison against the task-relevant GLUE-based measures.

### B.3.1 TASK-RELEVANT GEOMETRIC MEASURES FROM GLUE

To capture the aspects of representational geometry most relevant for classification, we employ the effective geometric measures introduced in the *Geometry Linked to Untangling Efficiency* (GLUE) framework (Chou et al., 2025a), grounded in manifold capacity theory (Chou et al., 2025a; Chung et al., 2018). The theory has found wide applications in both neuroscience (Yao et al., 2023; Paraouty et al., 2023; Kuoch et al., 2024; Hu et al., 2024) and machine learning (Cohen et al., 2020; Mamou et al., 2020; Stephenson et al., 2021; Kirsanov et al., 2025; Chou et al., 2025b).

Analogous to support vector machine (SVM) theory—where an analytical connection between the max-margin linear classifier and its support vectors is used to assess separability in the *best-case sense*—GLUE establishes a similar analytical connection in an *average-case sense*, as follows. Rather than analyzing the max-margin classifier directly in the original $N$-dimensional feature space $\mathbb{R}^N$, GLUE considers random projections to an $N'$-dimensional subspace and evaluates whether the representations remain linearly separable. Intuitively, if the data are highly separable in $\mathbb{R}^N$, they will, with high probability, remain separable even after projection to a much lower $N'$. Conversely, if the data are barely separable in $\mathbb{R}^N$, the probability of maintaining separability will rapidly drop to zero as $N'$ decreases.

Formally, following the modeling and notation in GLUE, each object manifold is modeled as the convex hull of all representations corresponding to the $\mu$-th class:

$$\mathcal{M}^\mu = \operatorname{conv}\left(\{\mathbf{z}_i^\mu\}_{i=1}^M\right),$$

where $\{\mathbf{z}_i^\mu\}$ is the collection of $M$ feature vectors of the $\mu$-th class. A dichotomy vector $\mathbf{y} \in \{-1, 1\}^P$ and a collection $\mathcal{Y} \subset \{-1, 1\}^P$ are chosen by the analyst. Common choices are $\mathcal{Y}$ being the set of all 1-vs-rest dichotomies (e.g., $(1, -1, -1, \ldots, -1)$, $(-1, 1, -1, \ldots, -1)$, ..., $(-1, -1, -1, \ldots, 1)$) or $\mathcal{Y} = \{-1, 1\}^P$.

The key quantity in GLUE for measuring the degree of (linear) separability of manifolds is the *critical dimension*, defined as the smallest $N'$ such that the probability of (linear) separability after projection to a random $N'$-dimensional subspace is at least $0.5$:

$$N_{\mathrm{crit}} := \min_{p(N') \geq 0.5} N',$$

where

$$p(N') := \Pr_{\Pi:\mathbb{R}^N \to \mathbb{R}^{N'}} \left[ \exists \, \mathbf{w} \in \mathbb{R}^{N'} \text{ s.t. } y^\mu \langle \mathbf{w}, \mathbf{x}^\mu \rangle \geq 0, \; \forall \mu, \; \mathbf{x}^\mu \in \mathcal{M}^\mu \right].$$

By scaling $N_{\text{crit}}$ with the number of manifolds, we define the *classification capacity* $\alpha := P/N_{\text{crit}}$, which intuitively captures the maximal load a network can handle. Larger $\alpha$ corresponds to more separable manifolds in the average-case sense.

GLUE theory relates $\alpha$ to manifold structure through:

$$\alpha = P \cdot \left( \mathbb{E}_{\substack{\mathbf{y} \sim \mathcal{Y} \\ \mathbf{t} \sim \mathcal{N}(0, I_N)}} \left[ \max_{\lambda_i^\mu \geq 0 \; \forall \mu, i} \left( \frac{\langle \mathbf{t}, \sum_{\mu, i} y^\mu \lambda_i^\mu \mathbf{z}_i^\mu \rangle}{\left\| \sum_{\mu, i} y^\mu \lambda_i^\mu \mathbf{z}_i^\mu \right\|_2} \right)^2 \right] \right)^{-1}. \tag{4}$$

Equation 4 can be numerically estimated using a quadratic programming solver (see Algorithm 1 in (Chou et al., 2025a)).

Observe that one can view the optimal solution $\lambda^\mu(\mathbf{y}, \mathbf{t})$ for the inner maximization problem as a function of $\mathbf{y}, \mathbf{t}$. This naturally leads to the following definition of *anchor point* for class $\mu$ as:

$$\mathbf{s}^\mu(\mathbf{y}, \mathbf{t}) := \frac{\sum_i \lambda_i^\mu(\mathbf{y}, \mathbf{t}) \mathbf{z}_i^\mu}{\sum_i \lambda_i^\mu(\mathbf{y}, \mathbf{t})},$$

and stacking them into a matrix $\mathbf{S} \in \mathbb{R}^{P \times N}$ and let $\mathbf{S_y} := \text{diag}(\mathbf{y}) \mathbf{S}$, GLUE yields an equivalent form:

$$\alpha = P \cdot \left( \mathbb{E}_{\substack{\mathbf{y} \sim \mathcal{Y} \\ \mathbf{t} \sim \mathcal{N}(0, I_N)}} \left[ (\mathbf{S_y t})^\top (\mathbf{S_y S_y}^\top)^\dagger (\mathbf{S_y t}) \right] \right)^{-1}, \tag{5}$$

where $\dagger$ denotes the pseudoinverse. This parallels SVM theory, where the margin is linked to a simple function on the support vectors.

**Center–axis decomposition of anchor points.** For each $\mu \in [P]$, define the anchor center of the $\mu$-th manifold as:

$$\mathbf{s}_0^\mu := \mathbb{E}_{\mathbf{y}, \mathbf{t}} \left[ \mathbf{s}^\mu(\mathbf{y}, \mathbf{t}) \right],$$

and for each $(\mathbf{y}, \mathbf{t})$, define the axis component of the $\mu$-th anchor point as:

$$\mathbf{s}_1^\mu(\mathbf{y}, \mathbf{t}) := \mathbf{s}^\mu(\mathbf{y}, \mathbf{t}) - \mathbf{s}_0^\mu.$$

Similar to $\mathbf{S_y}$, we denote $\mathbf{S}_{\mathbf{y}, 0}, \mathbf{S}_{\mathbf{y}, 1}(\mathbf{y}, \mathbf{t}) \in \mathbb{R}^{P \times N}$ as the matrices containing $y^\mu \mathbf{s}_0^\mu$ and $y^\mu \mathbf{s}_1^\mu(\mathbf{y}, \mathbf{t})$ on their rows, respectively, i.e., $\mathbf{S}_{\mathbf{y}, 0} := \text{diag}(\mathbf{y}) \mathbf{S}_0$ and $\mathbf{S}_{\mathbf{y}, 1}(\mathbf{y}, \mathbf{t}) := \text{diag}(\mathbf{y}) \mathbf{S}_1(\mathbf{y}, \mathbf{t})$ where $\mathbf{S}_0$ and $\mathbf{S}_1(\mathbf{y}, \mathbf{t})$ have $\mathbf{s}_0^\mu$ and $\mathbf{s}_1^\mu(\mathbf{y}, \mathbf{t})$ stacked on their rows.

With these, define:

$$a(\mathbf{y}, \mathbf{t}) = (\mathbf{S_y t})^\top (\mathbf{S_y S_y}^\top)^\dagger (\mathbf{S_y t}),$$

$$b(\mathbf{y}, \mathbf{t}) = (\mathbf{S}_{\mathbf{y}, 1}(\mathbf{y}, \mathbf{t}) \mathbf{t})^\top \left( \mathbf{S}_{\mathbf{y}, 1}(\mathbf{y}, \mathbf{t}) \mathbf{S}_{\mathbf{y}, 1}(\mathbf{y}, \mathbf{t})^\top \right)^\dagger (\mathbf{S}_{\mathbf{y}, 1}(\mathbf{y}, \mathbf{t}) \mathbf{t}),$$

$$c(\mathbf{y}, \mathbf{t}) = (\mathbf{S}_{\mathbf{y}, 1}(\mathbf{y}, \mathbf{t}) \mathbf{t})^\top \left( \mathbf{S}_{\mathbf{y}, 0} \mathbf{S}_{\mathbf{y}, 0}^\top + \mathbf{S}_{\mathbf{y}, 1}(\mathbf{y}, \mathbf{t}) \mathbf{S}_{\mathbf{y}, 1}(\mathbf{y}, \mathbf{t})^\top \right)^\dagger (\mathbf{S}_{\mathbf{y}, 1}(\mathbf{y}, \mathbf{t}) \mathbf{t}).$$

Note that $\alpha = P / \mathbb{E}_{\mathbf{y}, \mathbf{t}}[a(\mathbf{y}, \mathbf{t})]$.

**Effective geometric measures.** GLUE further decomposes $\alpha$ into three measures:

$$\alpha = \Psi_{\text{eff}} \cdot \frac{1 + R_{\text{eff}}^{-2}}{D_{\text{eff}}},$$

where:

- **Effective dimension:**

$$D_{\text{eff}} := \frac{1}{P} \, \mathbb{E}_{\mathbf{y}, \mathbf{t}}[b(\mathbf{y}, \mathbf{t})]$$

  Intuitively, $D_{\text{eff}}$ measures the intrinsic dimensionality of the manifolds while incorporating *axis alignment* between them. Lower $D_{\text{eff}}$ corresponds to more compact, better-aligned manifolds, improving linear separability.

- **Effective radius:**

$$R_{\text{eff}} := \sqrt{\frac{\mathbb{E}_{\mathbf{y},\mathbf{t}}[c(\mathbf{y},\mathbf{t})]}{\mathbb{E}_{\mathbf{y},\mathbf{t}}[b(\mathbf{y},\mathbf{t}) - c(\mathbf{y},\mathbf{t})]}}$$

Intuitively, $R_{\text{eff}}$ quantifies the scale of manifold variation relative to its center, incorporating *center alignment* between classes. Smaller $R_{\text{eff}}$ reflects tighter clustering of features around class centers, reducing manifold overlap.

- **Effective utility:**

$$\Psi_{\text{eff}} := \frac{\mathbb{E}_{\mathbf{y},\mathbf{t}}[c(\mathbf{y},\mathbf{t})]}{\mathbb{E}_{\mathbf{y},\mathbf{t}}[a(\mathbf{y},\mathbf{t})]}$$

Intuitively, $\Psi_{\text{eff}}$ measures the combined effect of *signal-to-noise ratio* (SNR) on separability. Higher $\Psi_{\text{eff}}$ corresponds to manifolds that are both low-dimensional and compact relative to inter-class distances.

For further derivations, illustrations, and examples, see the supplementary materials of (Chou et al., 2025a). In all our experiments, for each manifold we subsample to 50 points, conduct GLUE analysis on each manifold pair, and apply Gaussianization preprocessing (Wakhloo et al., 2023) to ensure initial linear separability.

$$\rho_{\mu,\nu}^{c} := |\langle \mathbf{s}_0^{\mu}, \mathbf{s}_0^{\nu} \rangle|$$

$$\rho_{\mu,\nu}^{a} := \mathbb{E}_{\mathbf{y},\mathbf{t}}\left[|\langle \mathbf{s}_1^{\mu}(\mathbf{y},\mathbf{t}), \mathbf{s}_1^{\nu}(\mathbf{y},\mathbf{t}) \rangle|\right]$$

$$\psi_{\mu,\nu} := \mathbb{E}_{\mathbf{y},\mathbf{t}}\left[|\langle \mathbf{s}_0^{\mu}, \mathbf{s}_1^{\nu}(\mathbf{y},\mathbf{t}) \rangle|\right]$$

**Implementation details.** In all our experiments, we consider the following specific hyperparameter choice for GLUE analysis. We randomly

**Intuitions for GLUE measures.** The three task-relevant geometric measures—$D_{\text{eff}}$, $R_{\text{eff}}$, and $\Psi_{\text{eff}}$—serve as markers that directly link geometric properties of object manifolds to classification efficiency. As we show in later sections, they are substantially more predictive of OOD performance than conventional measures. Here we summarize key properties, examples, and approximations of GLUE measures in Table 4 for intuition-building.

---

[13] For the $\mu$-th manifold, define its anchor center as $\mathbf{s}_0^{\mu} := \mathbb{E}_{\mathbf{t}}[\mathbf{s}^{\mu}(\mathbf{t})]$ and the axis-part of the anchor point as $\mathbf{s}_1^{\mu}(\mathbf{t}) := \mathbf{s}^{\mu}(t) - \mathbf{s}_0^{\mu}$. Intuitively, $\mathbf{s}_0^{\mu}$ is the mean representation for the $\mu$-class, and $\mathbf{s}_1^{\mu}(\mathbf{t})$ corresponds to the within-class variation/spread. $\langle \cdot, \cdot \rangle$ denotes inner product and $\| \cdot \|_2$ denotes $\ell_2$ norm. Formulas for uncorrelated random spheres provide a useful mental picture: $D_{\text{eff}}$ resembles the *Gaussian width*, equal to the sphere's dimension (Vershynin, 2018); $R_{\text{eff}}$ reflects the ratio of within-manifold variation to mean response; and $\Psi_{\text{eff}}$ corresponds to the fraction of error (i.e., inner product with $\mathbf{t}$) attributable to within-manifold variation.

[14] We follow a top-down view of feature learning (Chou et al., 2025b), where *features* are understood functionally through their consequences for computation (e.g., enabling linear separability) rather than as specific interpretable axes or neurons. This perspective emphasizes how representational geometry changes with feature usage without requiring explicit identification of the features themselves. Moreover, by thinking of a direction in the representation space as a feature (linear representation hypothesis (Park et al., 2024)), the effective geometric measures offer interpretation in feature learning as listed in the table.

Table 4: Intuitions for GLUE measures.

| | $D_{\text{eff}} \geq 0$ | $R_{\text{eff}} \geq 0$ | $\Psi_{\text{eff}} \in [0, 1]$ |
|---|---|---|---|
| Geometric intuition | Quantify the task-relevant dimensionality of object manifolds. | Quantify the task-relevant spread within each manifold relative to their centers. | Quantify the amount of excessive compression of untangling manifolds . |
| Effect on linear separability | More separable when $D_{\text{eff}}$ is small. | More separable when $R_{\text{eff}}$ is small. | More separable when $\Psi_{\text{eff}}$ is large. |
| Example | $D_{\text{eff}}$ equals the dimension of uncor. random spheres | $R_{\text{eff}}$ equals the radius of uncor. random spheres | Collapsing manifolds to points yields $\Psi_{\text{eff}} \to 0$. |
| Formula for uncorrelated random spheres[6] | $\frac{1}{P} \sum_\mu \mathbb{E}\left[\left(\frac{\langle \mathbf{s}_1^\mu(\mathbf{t}), \mathbf{t}\rangle}{\|\mathbf{s}_1^\mu(\mathbf{t})\|_2}\right)^2\right]$ | $\frac{1}{P} \sum_\mu \sqrt{\mathbb{E}\left[\left(\frac{\|\mathbf{s}_1^\mu(\mathbf{t})\|_2}{\|\mathbf{s}_0^\mu\|_2}\right)^2\right]}$ | $\frac{1}{P} \sum_\mu \mathbb{E}\left[\left(\frac{\langle \mathbf{s}_1^\mu(\mathbf{t}), \mathbf{t}\rangle}{\langle \mathbf{s}^\mu(\mathbf{t}), \mathbf{t}\rangle}\right)^2\right]$ |
| Interaction with correlations among manifolds | If within-manifold variations align along similar directions, $D_{\text{eff}}$ decreases. | If manifold centers move farther apart, $R_{\text{eff}}$ decreases. | If within-manifold variations reduce without improving separability, $\Psi_{\text{eff}}$ decreases. |
| Interpretation in feature learning[7] | Low $D_{\text{eff}}$ indicates a smaller set of feature modes in use. | Low $R_{\text{eff}}$ indicates more similar feature usage across examples within a class. | Low $\Psi_{\text{eff}}$ indicates inefficient compression of within-class variability. |

# C    ADDITIONAL RESULTS FOR SECTION 3

## C.1    IMPLEMENTATION DETAILS

During the initial exploration of how OOD performance varies across a wide range of final model states, we trained all architectures from scratch on CIFAR-10. We used two optimizers: SGD with a momentum of 0.9, and AdamW (Loshchilov & Hutter, 2019). We ran training for 200 epochs with a cosine annealing learning rate schedule, which smoothly decays the learning rate to zero, stabilizing late-stage representation geometry.

For each architecture and optimizer pair, we performed a systematic $4 \times 4$ grid search over the initial learning rate ($\eta_0$) and weight decay ($\lambda$). The specific values for each grid, which were tailored to each architecture family based on empirical best practices, are detailed in Table 5 and Table 6. This diverse grid was designed to produce models in various training regimes, from under- to over-regularized, allowing us to find cases where ID performance is stable while OOD performance varies — a key aspect of our analysis.

Table 5: Hyperparameter grid for SGD optimizer.

| Architecture | Initial learning rate list | Weight decay list |
|---|---|---|
| VGG (13/16/19) | [0.01000, 0.00333, 0.00111, 0.00037] | [0.0010000, 0.0003333, 0.0001111, 0.0000370] |
| ResNet (18/34/50) | [1.00000, 0.50000, 0.25000, 0.12500] | [0.0002000, 0.0001000, 0.0000500, 0.0000250] |
| DenseNet121 | [0.05000, 0.01667, 0.00556, 0.00185] | [0.0005000, 0.0001667, 0.0000556, 0.0000185] |
| MobileNet | [0.20000, 0.06667, 0.02222, 0.00741] | [0.0001000, 0.0000333, 0.0000111, 0.0000037] |
| EfficientNetB0 | [0.20000, 0.06667, 0.02222, 0.00741] | [0.0001000, 0.0000333, 0.0000111, 0.0000037] |

Table 6: Hyperparameter grid for AdamW optimizer.

| Architecture | Initial learning rate list | Weight decay list |
|---|---|---|
| VGG (13/16/19) | [0.02000, 0.00500, 0.00125, 0.00031] | [0.0100000, 0.0033333, 0.0011111, 0] |
| ResNet (18/34/50) | [0.10000, 0.02500, 0.00625, 0.00156] | [0.0100000, 0.0050000, 0.0025000, 0] |
| DenseNet121 | [0.05000, 0.02500, 0.01250, 0.00625] | [0.0100000, 0.0033333, 0.0011111, 0] |
| MobileNet | [0.02000, 0.00500, 0.00125, 0.04000] | [0.0100000, 0.0033333, 0.0011111, 0] |
| EfficientNetB0 | [0.10000, 0.05000, 0.01000, 0.00100] | [0.0010000, 0.0003333, 0.0001111, 0] |

## C.2    DETAILS FOR FIGURE 4

In this section, we present supplementary figures that provide a more detailed view of the main findings stated in Figure 4 from Section 3. Table 7 provides a list of content for this subsection.

**Quantification of Relationships.**    We quantify the relationship between ID measures and OOD performance by computing the Pearson correlation coefficient ($r$) and its associated $p$-value via ordinary least-squares linear regression between the measure values and OOD accuracies. For all figures with heatmaps, we annotate each $r$-value with significance asterisks based on its $p$-value: $p \leq 0.0001$ (****), $p \leq 0.001$ (***), $p \leq 0.01$ (**), and $p \leq 0.05$ (*).

**Data Splits for Measures.**    The terms "Test" and "Train" in the figure labels indicate whether the representational measures were computed on the ID test set or the ID training set, respectively.

Table 7: Organization of figures in Appendix C.

| Figure label | Model set | Optimizer | ID split | OOD dataset |
|---|---|---|---|---|
| Figure 7 | Five DNNs | SGD | Train | CIFAR-100 |
| Figure 8 | Five DNNs | SGD | Test | CIFAR-100 |
| Figure 9 | Five DNNs | AdamW | Train | CIFAR-100 |
| Figure 10 | Five DNNs | AdamW | Test | CIFAR-100 |
| Figure 11 | Three ResNets + Three VGGs | SGD | Train | CIFAR-100 |
| Figure 12 | Three ResNets + Three VGGs | SGD | Test | CIFAR-100 |
| Figure 13 | ResNet18 + VGG19 | SGD | Train | ImageNet subset |
| Figure 14 | ResNet18 + VGG19 | SGD | Test | ImageNet subset |

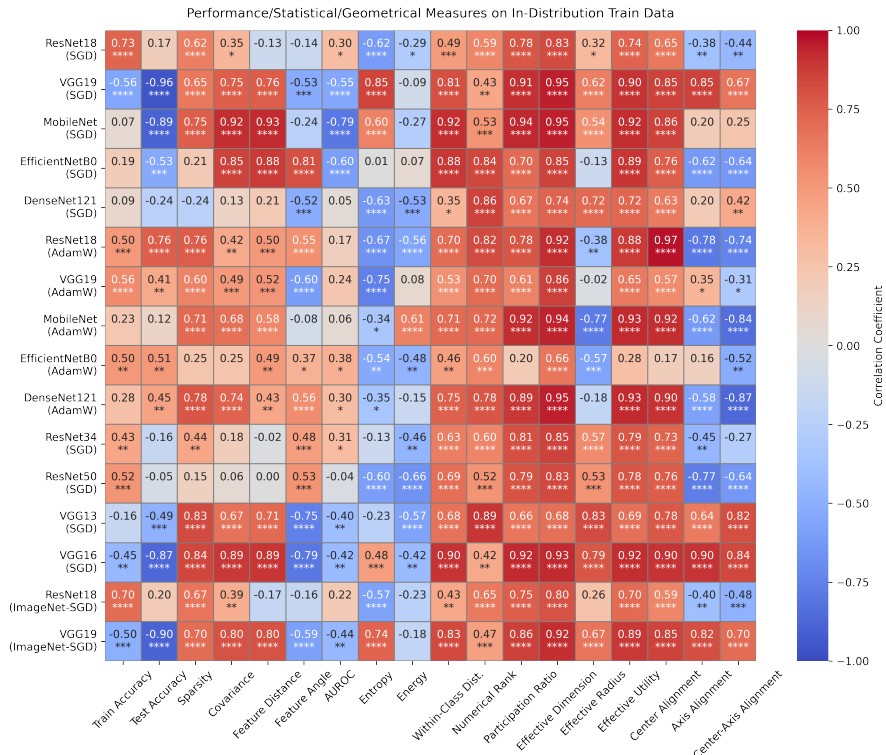

Figure 6: All results, measures computed on the ID *train* set.

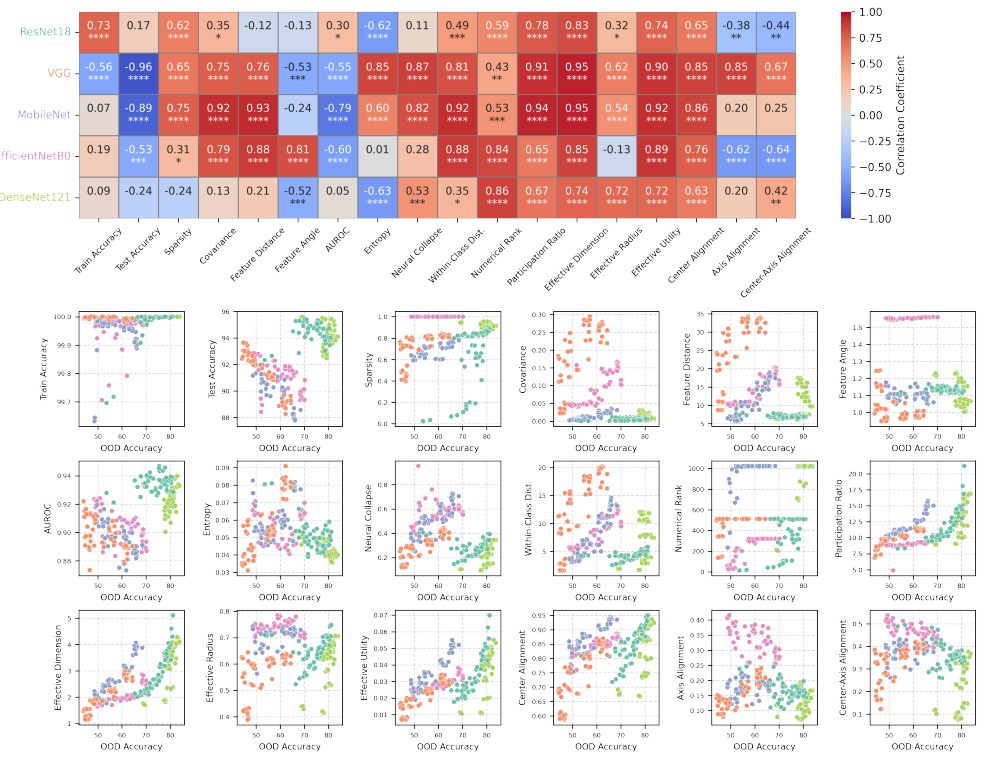

Figure 7: Five DNN architectures, trained with SGD, measures computed on the ID *train* set.

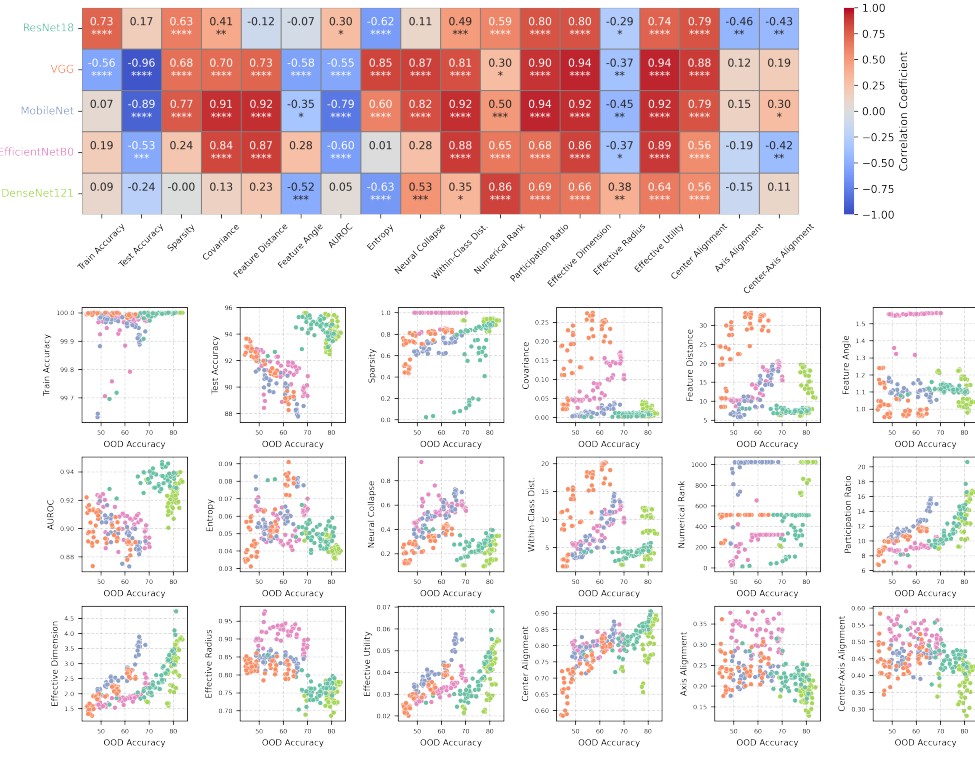

Figure 8: Five DNN architectures, trained with SGD, measures computed on the ID *test* set.

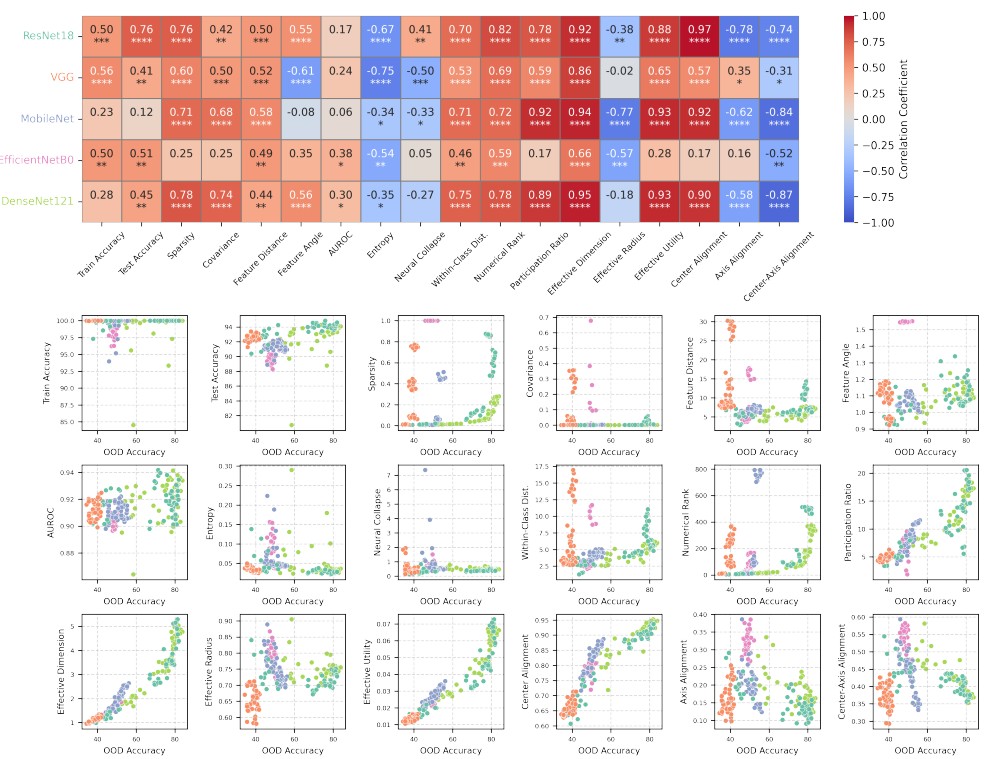

Figure 9: Five DNN architectures, trained with AdamW, measures computed on the ID *train* set.

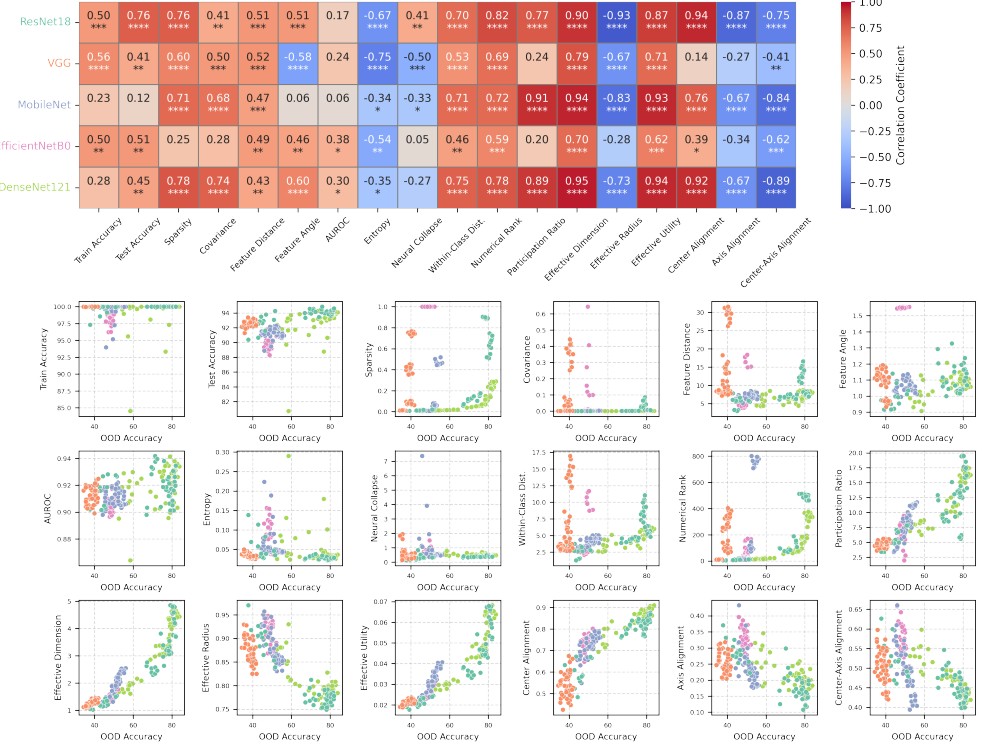

Figure 10: Five DNN architectures, trained with AdamW, measures computed on the ID *test* set.

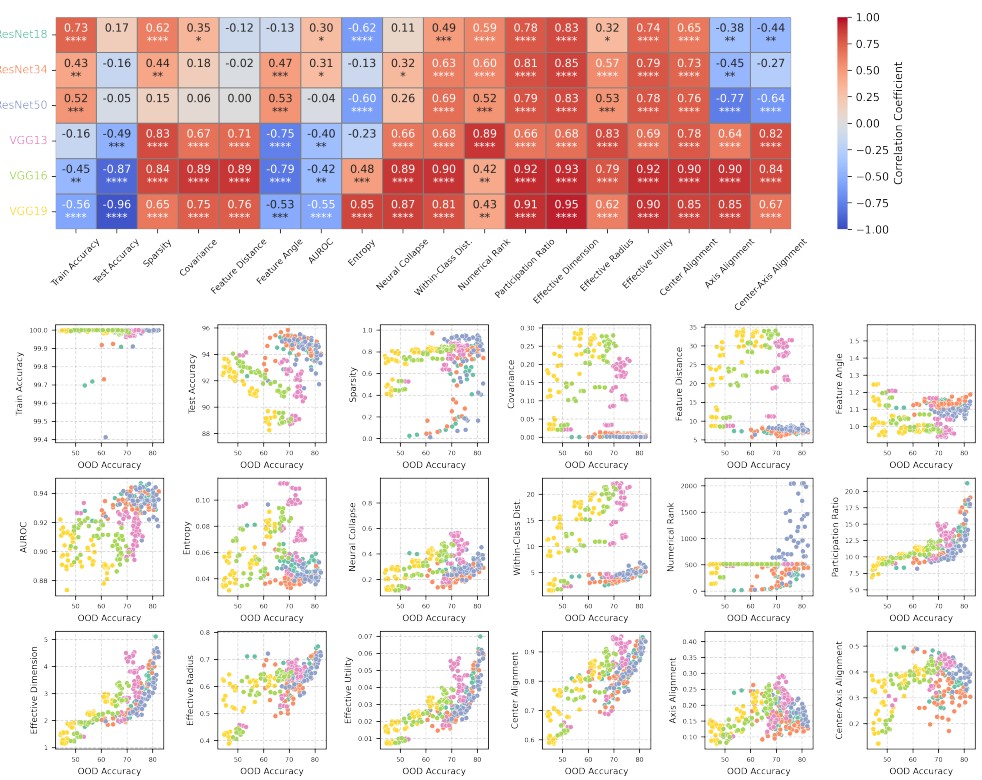

Figure 11: Three ResNet and three VGG architectures, trained with SGD, measures computed on the ID *train* set.

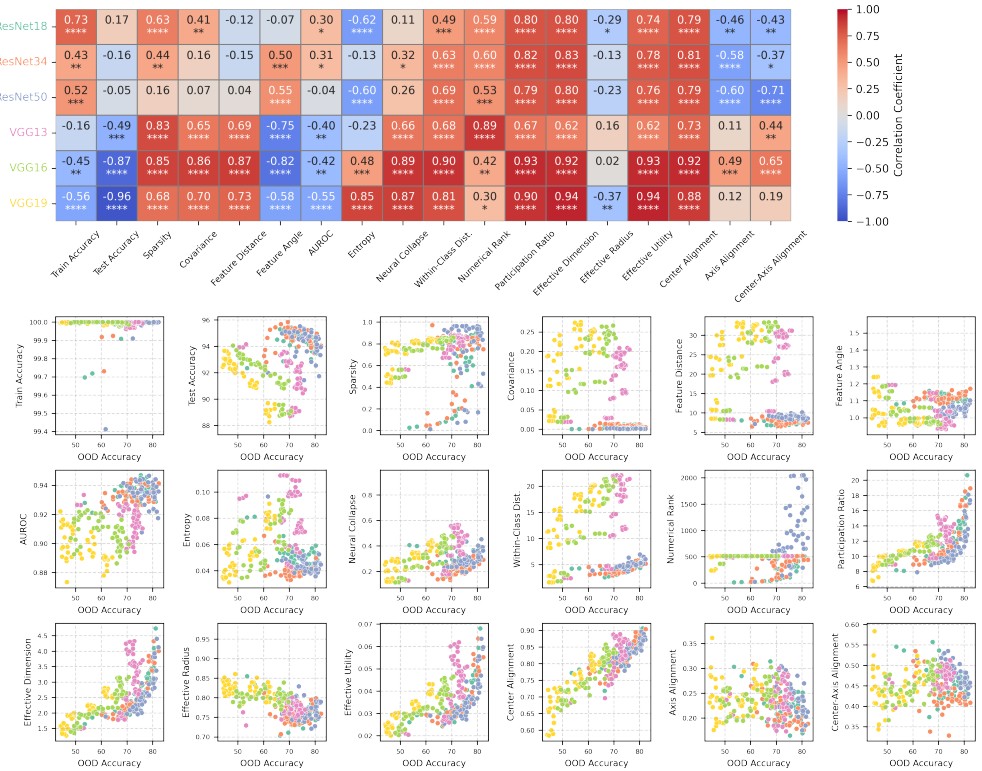

Figure 12: Three ResNet and three VGG architectures, trained with SGD, measures computed on the ID *test* set.

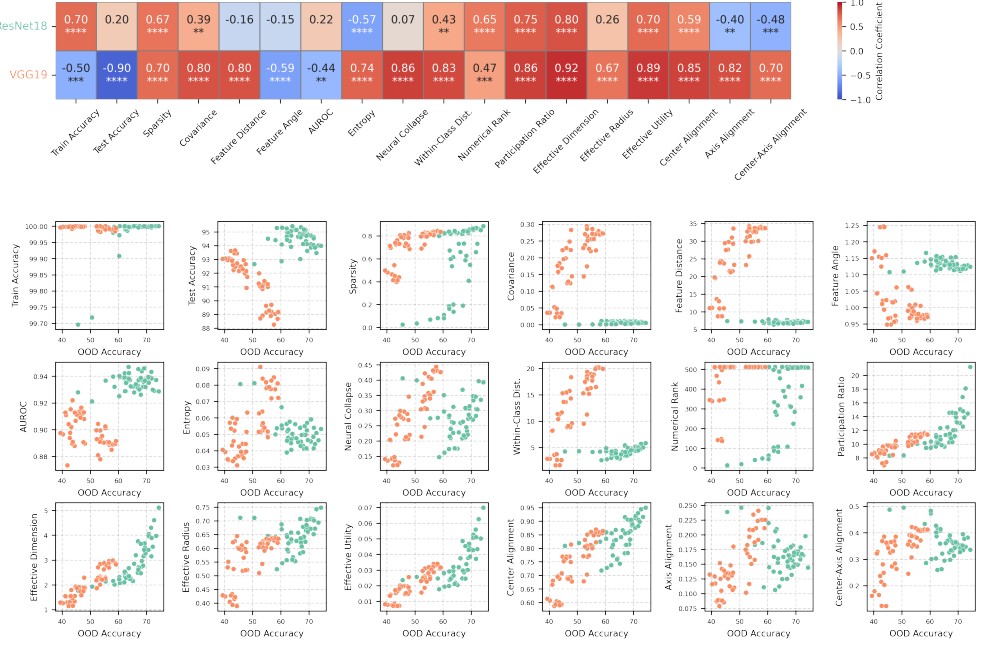

Figure 13: ResNet18 and VGG19, trained with SGD, evaluated on ImageNet subset OOD, measures computed on the ID *train* set.

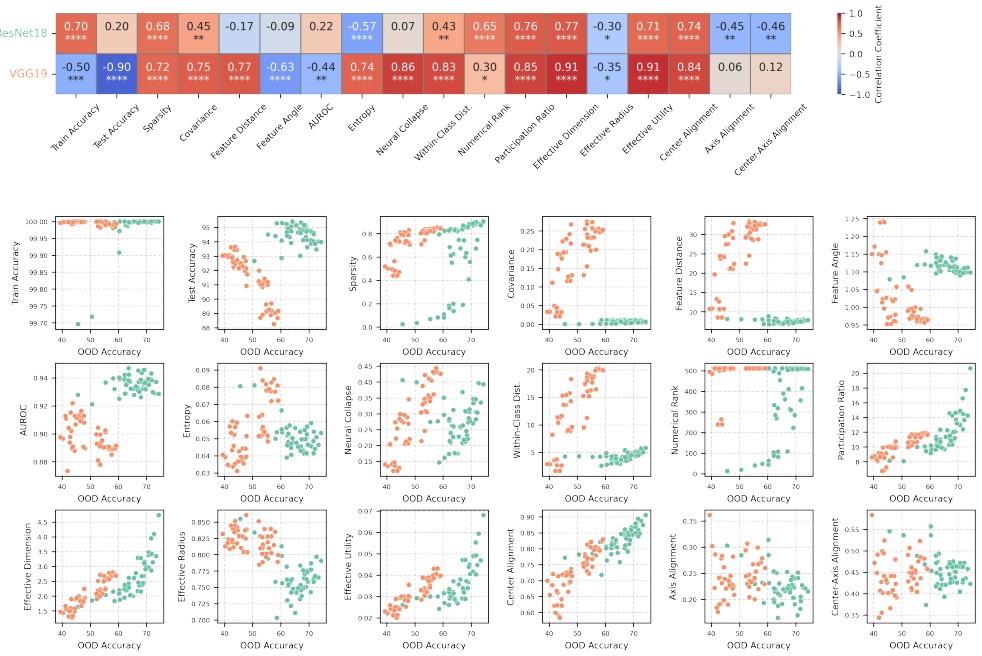

Figure 14: ResNet18 and VGG19, trained with SGD, evaluated on ImageNet subset OOD, measures computed on the ID *test* set.

### C.3 RESULTS ON CORRUPTED IMAGES AS OOD DATA

Here we provide results on 6 out of 19 corruption methods in CIFAR-10C.

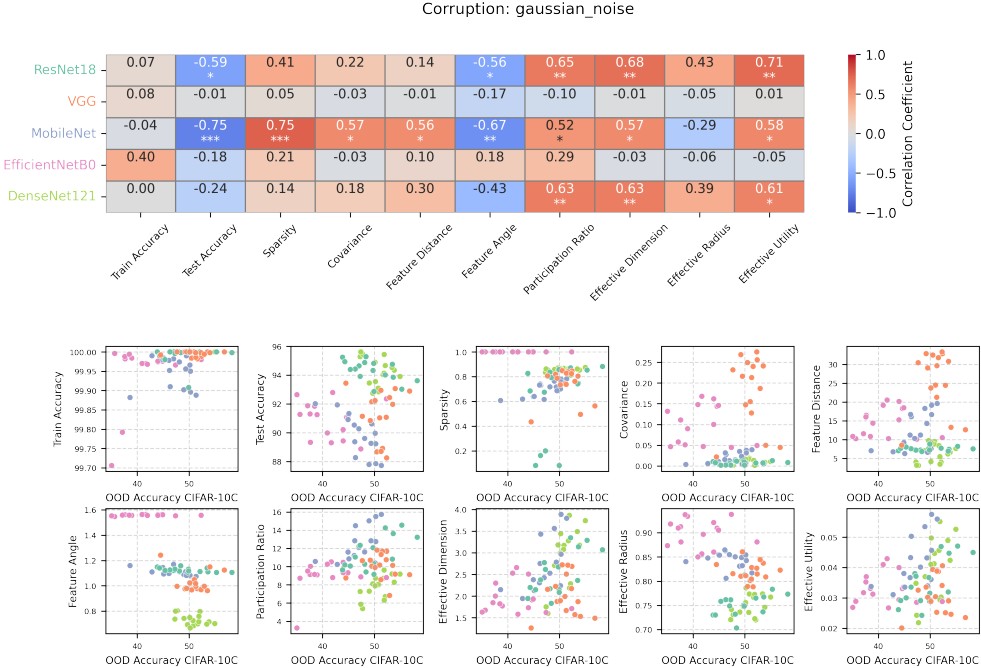

Figure 15: Corruption type: gaussian noise. Five DNN architectures, trained with SGD, measures computed on the ID *test* set.

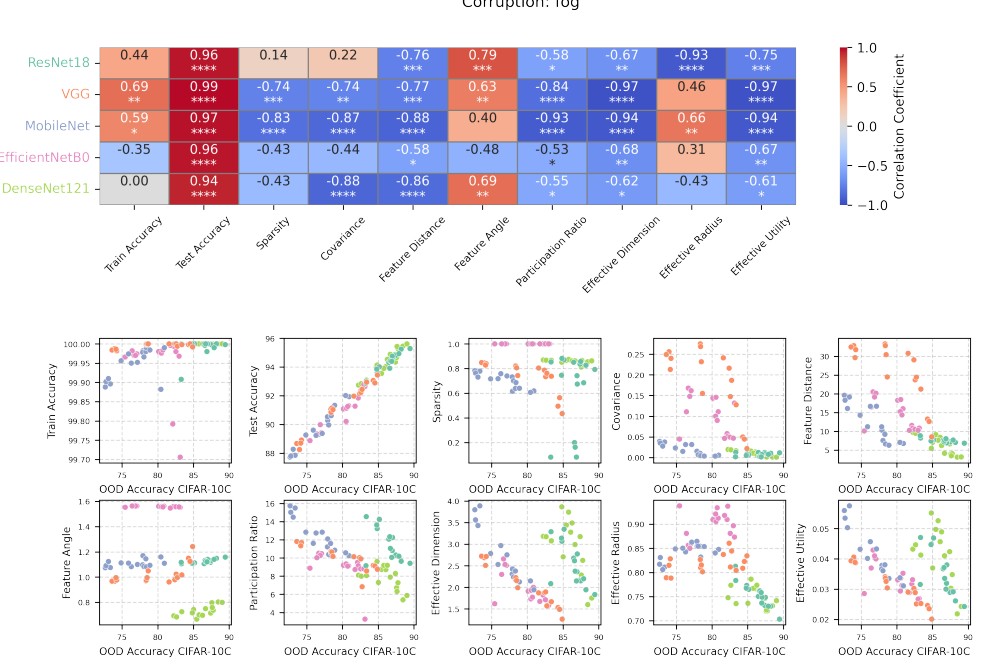

Figure 16: Corruption type: fog. Five DNN architectures, trained with SGD, measures computed on the ID *test* set.

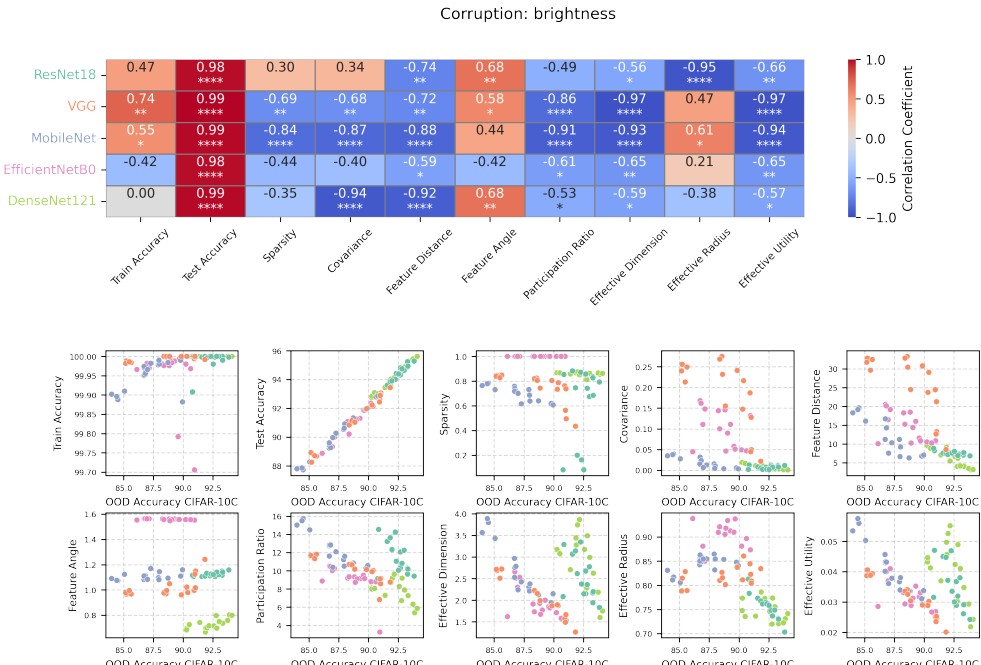

Figure 17: Corruption type: brightness. Five DNN architectures, trained with SGD, measures computed on the ID *test* set.

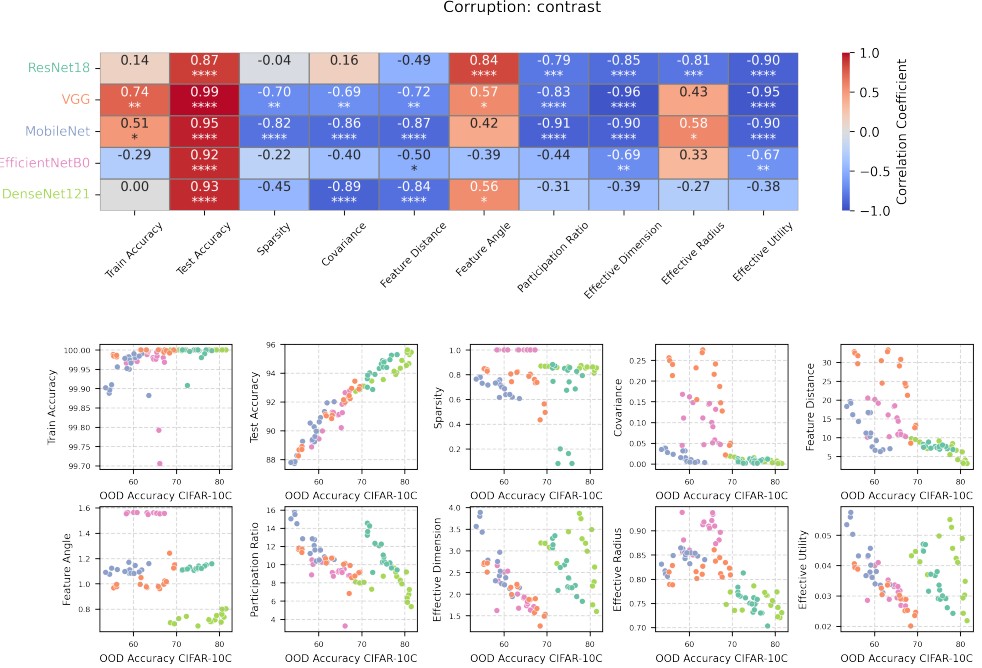

Figure 18: Corruption type: contrast. Five DNN architectures, trained with SGD, measures computed on the ID *test* set.

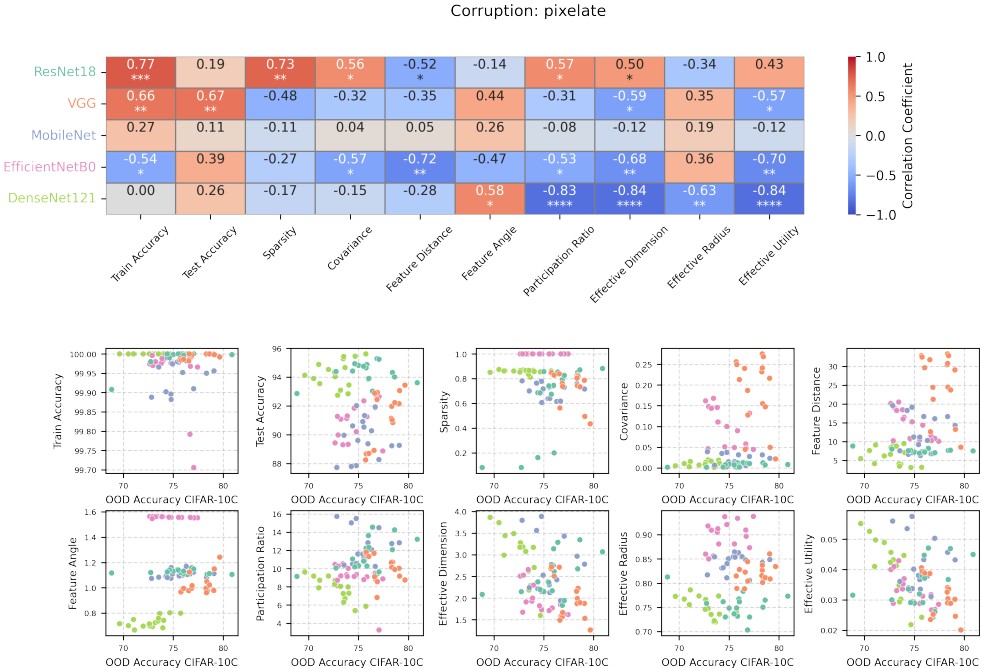

Figure 19: Corruption type: pixelate. Five DNN architectures, trained with SGD, measures computed on the ID *test* set.

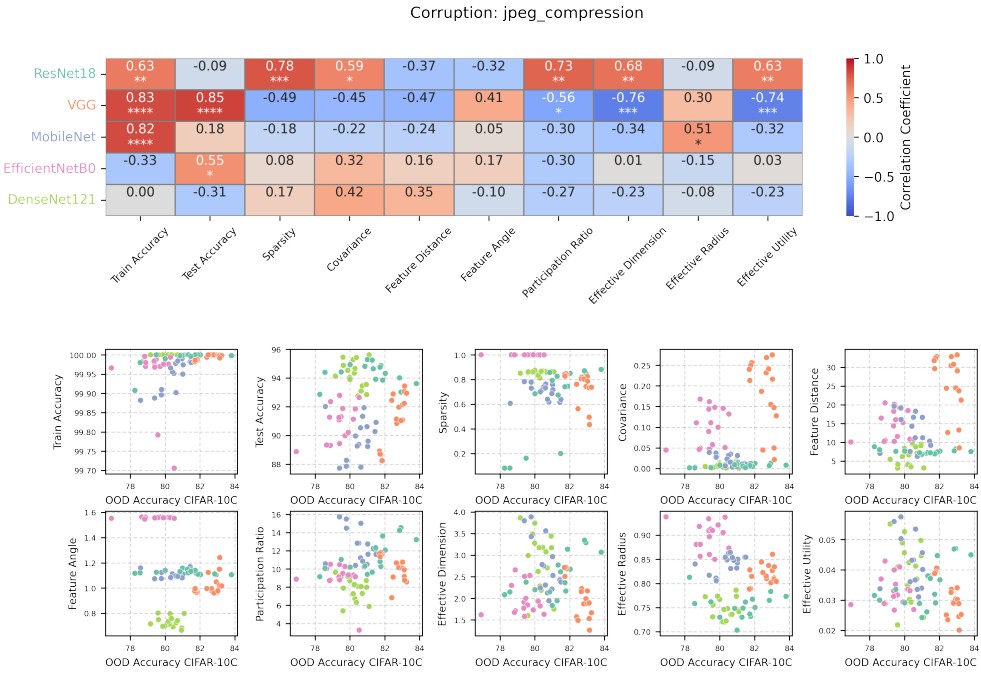

Figure 20: Corruption type: jpeg compression. Five DNN architectures, trained with SGD, measures computed on the ID *test* set.

# D  DETAILS ON THE APPLICATIONS TO PRETRAINED MODELS

Here we provide implementation details and statistical procedures underlying the pretrained model analysis in Section 4. This section accompanies the full results reported in Figure 21 and Figures 22, 23.

## D.1  MODEL SELECTION AND WEIGHTS

We evaluated 20 pretrained architectures available through the PyTorch model zoo, spanning families such as RegNet, MobileNet, ResNet/ResNeXt, WideResNet, EfficientNet, and Vision Transformer (ViT). For each architecture, we included both the `v1` and `v2` weight releases. The two weight sets differ in training recipes and regularization schemes, though exact details are not always disclosed, making them a heterogeneous and realistic testbed. By design, the `v2` models typically achieve higher ImageNet top-1 accuracy, while `v1` weights often exhibit higher manifold dimensionality.

We remark that for the ViT models, we treat `IMAGENET1K_SWAG_LINEAR_V1` as `v1` and `IMAGENET1K_SWAG_E2E_V1` as `v2`.

## D.2  REPRESENTATION EXTRACTION

For each model, we extracted feature representations from the penultimate layer (see Table 3 for exact layer names). Input images were preprocessed by resizing to $224 \times 224$ pixels, converted to tensors, and normalized with standard ImageNet statistics. For GLUE analysis, we subsampled 2 classes and for each class we subsampled 50 feature vectors, applied Gaussianization preprocessing, and computed effective geometric measures ($D_{\text{eff}}, R_{\text{eff}}, \Psi_{\text{eff}}$) as described in Appendix B. We repeated the above random subsampling for 100 times.

## D.3  OOD EVALUATION VIA LINEAR PROBING

To evaluate the OOD generalization of the frozen feature extractor, we attached a linear classifier to the penultimate feature representation of each pretrained model (see Table 3 for layer details). Crucially, the pretrained backbone weights remained frozen throughout this process; only the parameters of the new classifier were trained. For each OOD dataset, we train linear classifiers on the penultimate feature vectors for 50 epochs using the Adam optimizer with an initial learning rate of 0.1 and a cross-entropy loss function. In all the results, we report the average linear probe accuracy over 3 repetitions on different random seeds.

## D.4  PROGNOSTIC PREDICTION

For each model, after measuring the ($D_{\text{eff}}, \Psi_{\text{eff}}$) of `v1` and `v2` respectively. We use the following criteria to make a prognostic prediction: if the $D_{\text{eff}}(x) - D_{\text{eff}}(y)$ is greater than the sum of the standard error of estimating $D_{\text{eff}}(x)$ and $D_{\text{eff}}(y)$, plus $\Psi_{\text{eff}}(x) - \Psi_{\text{eff}}(y)$ is greater than the sum of the standard error of estimating $\Psi_{\text{eff}}(x)$ and $\Psi_{\text{eff}}(y)$, then we predict $x$ is going to have better OOD performance than $y$; otherwise we make no verdict (here $x, y \in \{\texttt{v1}, \texttt{v2}\}$).

Recall that in Section 4 we applied our prognostic method to 20 ImageNet-pretrained models across 9 OOD datasets and achieved a prediction accuracy of 73.02% (compared to 37.22% when using ID test accuracy as the marker). Here, we systematically evaluate other markers that showed reasonable performance in Section 3.2. Specifically, we consider $D_{\text{eff}}$ and $\Psi_{\text{eff}}$ as before, along with the Neural Collapse metric, numerical rank, average within-class distance, and participation ratio (definitions in Appendix B).

The prediction procedure follows the same criterion described earlier: for each marker, we compare the two weight versions (`v1` vs. `v2`) and issue a prediction only when the gap between their marker values exceeds the sum of the standard errors of estimation. We evaluate both individual markers and pairwise combinations.

The results, summarized in Figure 21, show that all these markers substantially outperform ID test accuracy as prognostic indicators of OOD transfer performance.

**Remark on alternative markers and future directions.** As shown in Fig. 21, several alternative markers—or combinations of markers—also achieve strong prognostic performance, and in some cases perform comparably to or slightly better than the specific pair $(D_{\mathrm{eff}}, \Psi_{\mathrm{eff}})$ used in the main analysis. This is fully consistent with the broader message of our work: a wide range of manifold-geometry-based quantities, both within and outside the GLUE family, contain significant predictive signal for OOD transfer performance. A deeper understanding of why different markers succeed on different subsets of architectures and how these markers may complement one another is an exciting direction for future investigation.

It is important to emphasize that the goal of the present experiment is not to identify a single "optimal" marker, but rather to demonstrate that geometric markers offer a substantial improvement over the conventional practice of using ID test accuracy as a predictor of OOD performance. Indeed, across all markers and marker-pairs we evaluated, the resulting prediction accuracies (ranging from 62% to 76%) consistently exceed that of ID test accuracy (37.22%) by a factor of approximately two. This reinforces the central conclusion that geometry-based diagnostics provide a robust and broadly effective alternative for prognostic prediction in transfer learning.

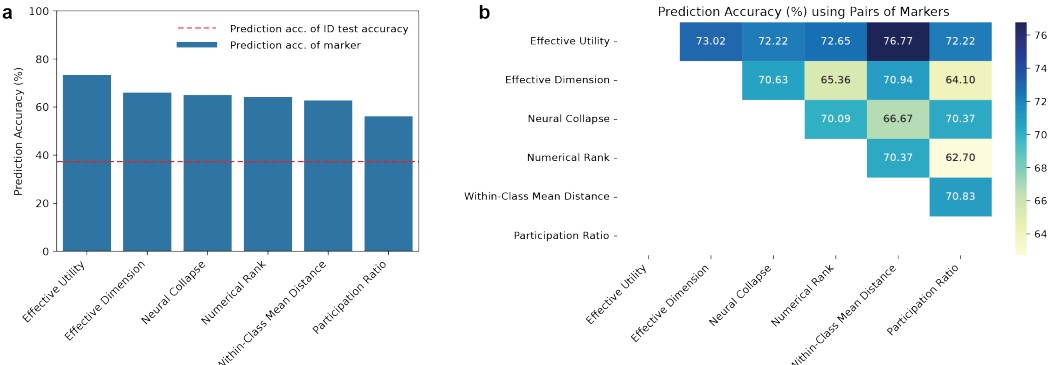

Figure 21: Prediction accuracy of OOD performance using different markers (or marker combinations). **a**, Using single marker. **b**, Using a pair of markers.

### D.5 FULL MODEL FINE-TUNING PROTOCOL

As a complementary evaluation, we also performed end-to-end fine-tuning. Models were initialized with either the v1 or v2 pretrained weights, and a new task-specific classifier head was randomly initialized. Unlike the linear probe, all model parameters (both in the backbone and the new classifier) were updated during training.

To simulate a realistic application scenario, we fine-tuned the models on the complete official training splits of Flowers102 (6,149 images) and Stanford Cars (8,144 images). Training was conducted for 50 epochs with a batch size of 64. We used the AdamW optimizer (Loshchilov & Hutter, 2019) with a weight decay of $10^{-6}$ and a cosine annealing learning rate scheduler with an initial learning rate of $3 \times 10^{-4}$.

To monitor the learning dynamics, we evaluated the model's performance on the validation set at 40 checkpoints, spaced logarithmically throughout the training process.

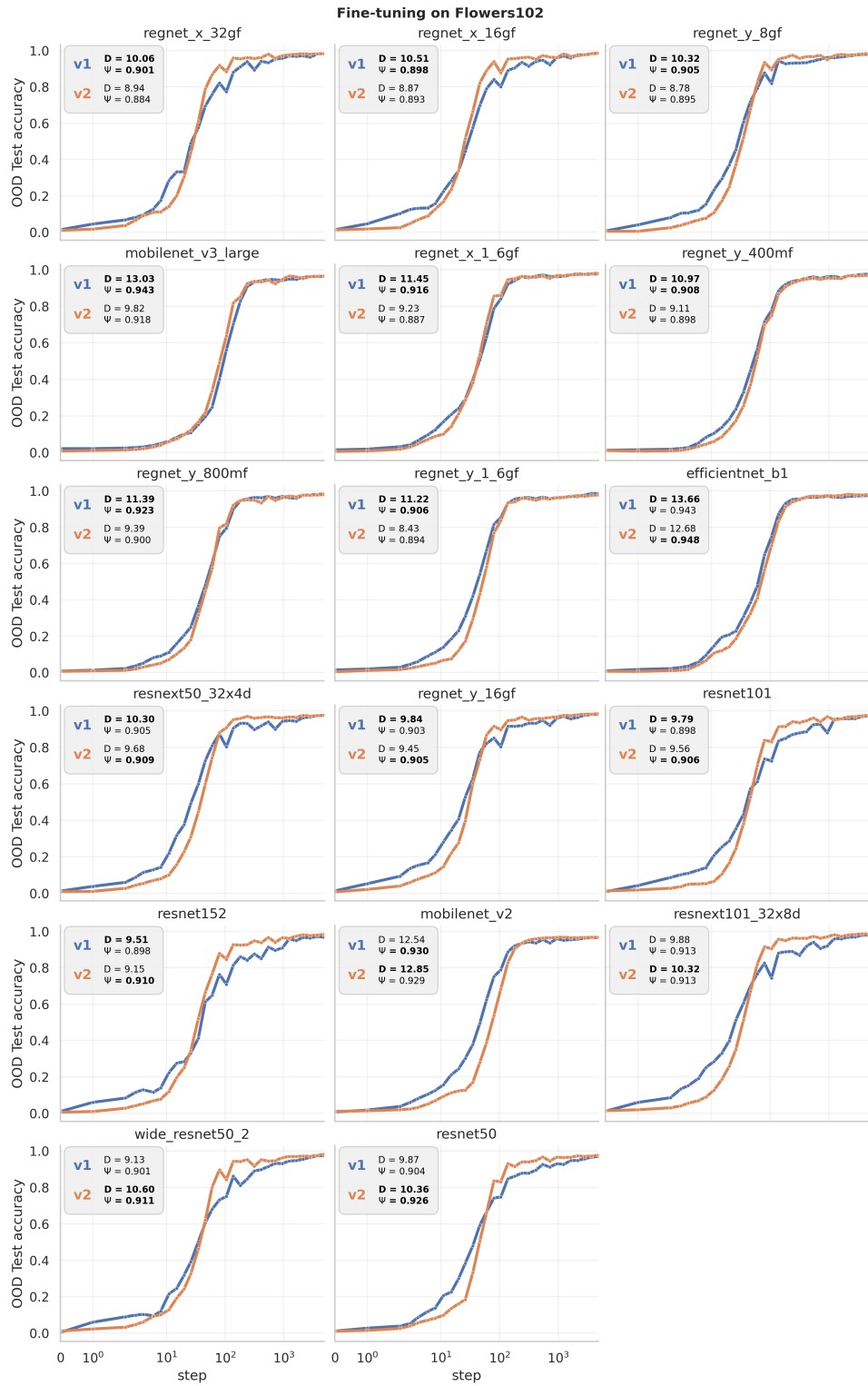

Figure 22: Fine-tuning dynamics of ImageNet-pretrained networks on **Flowers102** dataset from v1 and v2 weights. Insets show ID measures at initialization

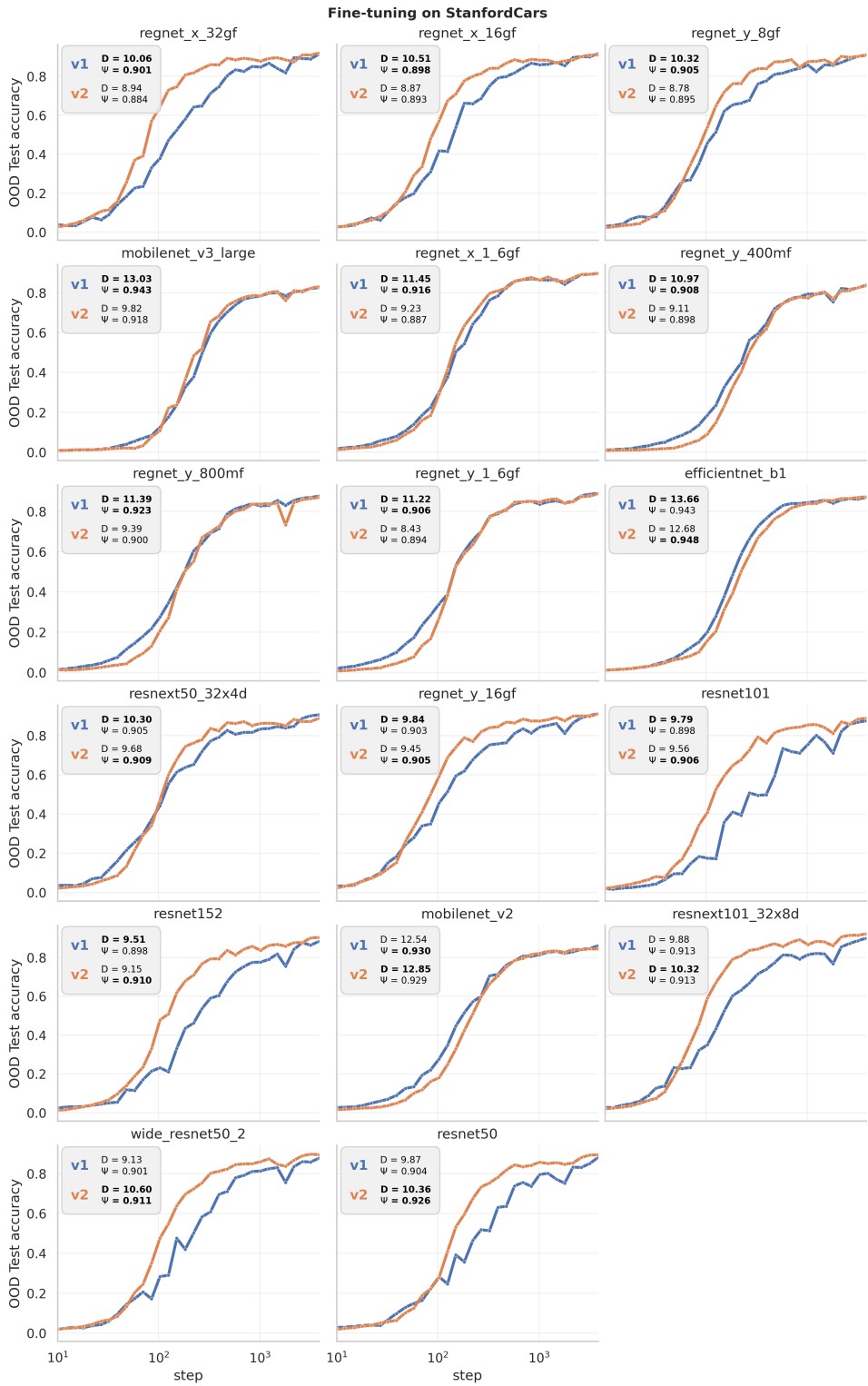

Figure 23: Fine-tuning dynamics of ImageNet-pretrained networks on **StanfordCars** dataset from v1 and v2 weights. Insets show ID measures at initialization

