# OpenReview forum: "Diagnosing Generalization Failures from Representational Geometry Markers"
_ICLR.cc/2026/Conference — ICLR 2026 Poster_

### Official Review · Reviewer_YFoW · 2025-10-31

**Soundness:** 3
**Presentation:** 3
**Contribution:** 3
**Rating:** 6
**Confidence:** 3

**Summary:**

The paper proposes to predict when models will fail to generalize especially under distribution shifts, even before deploying them. A diagnostic framework is presented that uses measurable properties of the model's internal representations. Through experiments on CIFAR-10/100 and ImageNet, the authors show that simple geometric indicators of in-distribution (ID) features, such as effective manifold dimension (Deff) and effective utility (Psi_eff), can serve as biomarkers for out-of-distribution (OOD) robustness. Applied to ImageNet-pretrained architectures, these metrics are able to identify which model weights will transfer better to new datasets like Flowers, Cars, and Places.

**Strengths:**

The paper addresses a real gap: predicting generalization failure before deployment.

The results hold across several architectures, optimizers, and datasets.

Although dense, the paper is well-written and the figures are quite helpful.

**Weaknesses:**

While the methods are well explained, the paper could present a step-by-step practical guide (an actual "diagnotic kit", as the authors suggest themselves). This would make the findings more easy for practitioners to apply when designing new architectures.

It is unclear how this framework generalizes to language or multimodal models. A short discussion about this or a pilot result would considerably strengthen the relevance of these results.

**Questions:**

How computationally expensive are the GLUE metrics used in this paper?

What hyperparameters (if any) that must be set for computing the GLUE metrics?

In practice, are their any thresholds of when Deff or Psi_eff would indicate "good" vs. "bad" geometry?

---

> ### Author Response · Authors · 2025-11-20
> **Response part 1**
>
> We thank the reviewer for the thoughtful and constructive feedback. We appreciate the recognition that (i) predicting OOD failure *before deployment* is an important and underexplored problem, (ii) our empirical evaluations are consistent across architectures and datasets, and (iii) the geometric markers D_eff and Psi_eff provide practically useful insights for model selection. The reviewer’s comments helped us clarify both practical aspects of our diagnostic pipeline and the broader applicability of the framework.
>
> In the revision, and in response to this reviewer and others, we have added or expanded several components:
>
> - **A clearer explanation of the diagnostic framework**, emphasizing how it differs from performance-driven paradigms and how it can be operationalized in practice(**idQw, H4Eg, YFoW**).
> - Additional comparison with existing OOD detection methods [1-5] and transfer learning studies [6-8]. Moreover, we implemented markers based on these methods, including logit entropy [12], AUROC [11] neural collapse metrics [8,9], and tunnel effect metrics [6,7] (**idQw, H4Eg**).
> - **Improved explanation of the computational cost and stability** of GLUE metrics (**H4Eg, YFoW**).
> - **Discussing extensions to language and multimodal models**, focusing on how to define failure modes and markers for those domains (**idQw, H4Eg, YFoW**).
>
> We hope these clarifications address the reviewer’s concerns and further strengthen the paper.
>
> ## **Detailed Response**
>
> ### **1) Practical guide / “diagnostic kit”**
>
> We thank the reviewer for this suggestion.
>
> **Reviewer YFoW**: “While the methods are well explained, the paper could present a step-by-step practical guide (an actual "diagnotic kit", as the authors suggest themselves). This would make the findings more easy for practitioners to apply when designing new architectures.”
>
> We agree that diagnostic reasoning is most useful when paired with a clear, reproducible workflow. In the revised version, we explicitly add:
>
> - a reference to the complete implementation in the supplementary materials (Appendix B), and
> - a clarification of how this contrasts with the standard performance-driven ML paradigm (Line 162-185).
>
> We also note that the *structure* of a diagnostic kit depends on the task: while image classification allows a relatively simple and uniform pipeline, more complex tasks (e.g., hierarchical tasks, RL, continuous-control problems) may require multi-level markers aligned to the structure of the task. We emphasize this conceptual distinction to clarify what “diagnostic” means in our framework.
>
> ### **2) Generalizing beyond image classification (language / multimodal)**
>
> We agree this is an important direction and appreciate the chance to clarify.
>
> **Reviewer YFoW**: “It is unclear how this framework generalizes to language or multimodal models. A short discussion about this or a pilot result would considerably strengthen the relevance of these results.”
>
> In the revision, we add a dedicated discussion section outlining:
>
> - The first step is to **identify failure modes** in the target domain (e.g., hallucination for LLMs, chain-of-thought inconsistencies, and multimodal grounding errors).
> - Once failure modes are clearly defined, the same diagnostic philosophy applies: systematically compare models that differ in failure rate but have similar ID performance, and analyze their internal representations (token embeddings, sequence embeddings, multimodal fusion layers, decoder states, etc.) to identify predictive markers.
> - We emphasize that GLUE is one specific instantiation for classification manifolds, and analogous task-relevant measures can be derived for language or multimodal settings.
>
> This helps articulate how the framework generalizes without requiring new experiments in this revision.
>
> ### **3) Computational cost of GLUE metrics**
>
> We clarify in the revision that **the dominant cost is in computing linear-probe performance**, not in estimating GLUE metrics. The core computation in GLUE is solving quadratic programs with linear constraints. And the size of the problem is the number of features by the number of total points (i.e., the number of classes times the number of samples per class). As we downsampled the number of points per class to 50 in all our GLUE analyses and study pairwise manifold geometry, the size of the problem is #features x 100. And empirically, on a single CPU, one run of GLUE analysis can be done in 30 seconds.

---

> > ### Author Response · Authors · 2025-11-20
> > **Response part 2**
> >
> > ### **4) Hyperparameters of GLUE metrics**
> >
> > The GLUE markers use only a small number of stable hyperparameters:
> >
> > - subsampling of examples within each class.
> > - pairwise manifold classification
> >
> > These choices are described in the supplementary materials. Empirically, we find the metrics to be **insensitive** to these values over wide ranges, consistent with previous analyses in the GLUE literature. The revision adds a concise description of these settings to help practitioners reproduce results.
> >
> > ### **5) Thresholds for D_eff and Psi_eff**
> >
> > We appreciate the reviewer’s question and agree this deserves clarification.
> >
> > **Reviewer YFoW**: “In practice, are their any thresholds of when Deff or Psi_eff would indicate "good" vs. "bad" geometry?”
> >
> > At present, absolute thresholds are task- and model-dependent, so the most reliable use case is relative comparison:
> >
> > - when ID performance is similar, higher D_eff and higher Psi_eff generally indicate more robust, less overspecialized representations;
> > - unusually low values can signal overcompression or shortcut-like behavior;
> > - unusually high values suggest undertrained or noisy representations.
> >
> > For image classification, we emphasize in the revision that **the markers are best used to compare nearby models** (e.g., same family, different training recipes). Developing universal thresholds is a promising direction, but beyond the scope of the present work.
> >
> > ## References
> >
> > [1] Distribution Shift Inversion for Out-of-Distribution Prediction: https://arxiv.org/abs/2306.08328
> >
> > [2] Predicting Out-of-Distribution Error with the Projection Norm: https://proceedings.mlr.press/v162/yu22i.html
> >
> > [3] To Annotate or Not? Predicting Performance Drop under Domain Shift: https://aclanthology.org/D19-1222/
> >
> > [4] Can you trust your model's uncertainty? Evaluating predictive uncertainty under dataset shift: https://arxiv.org/abs/1906.02530
> >
> > [5] Unpacking Softmax: How Temperature Drives Representation Collapse, Compression, and Generalization, https://arxiv.org/abs/2506.01562
> >
> > [6] The Tunnel Effect: Building Data Representations in Deep Neural Networks: https://arxiv.org/abs/2305.19753
> >
> > [7] What Variables Affect Out-of-Distribution Generalization in Pretrained Models?: https://arxiv.org/abs/2405.15018
> >
> > [8] Controlling Neural Collapse Enhances Out-of-Distribution Detection and Transfer Learning: [https://arxiv.org/pdf/2502.1091](https://arxiv.org/pdf/2502.10691)
> >
> > [9] NECO: NEURAL COLLAPSE BASED OUT-OF-DISTRIBUTION DETECTION, https://arxiv.org/pdf/2310.06823
> >
> > [10] Delays in generalization match delayed changes in representational geometry: https://proceedings.mlr.press/v285/zheng24a.html
> >
> > [11] Benchmarking Neural Network Robustness to Common Corruptions and Perturbations: https://arxiv.org/abs/1903.12261
> >
> > [12] Predicting With Confidence on Unseen Distributions: https://arxiv.org/abs/2107.03315

---

> ### Comment · Reviewer_YFoW · 2025-11-26
> **Response to rebuttal**
>
> I thank the authors for their careful responses and important  additions to the manuscript.
>
> __1) Practical guide / “diagnostic kit”__
>
> _"... a reference to the complete implementation in the supplementary materials (Appendix B), ..."_
>
> I believe the authors mean Appendix D. What was added is certainly helpful. For example, the fact that they compare the difference in GLUE metrics between the two weight sets (v1 and v2) against the sum of the metrics' standard errors was nowhere mentioned in the manuscript before.
>
> Now that I better understand your method, several questions arise:
>
> - How is the standard error of estimating D_eff and Psi_eff measured? Can the authors please include that information in the Appendix?
>
> - Are the marker values in Fig. 5 being reported as averages? If yes, over how many runs?
>
> - In Fig. 4, why is D_eff negatively correlated with OOD performance? Don't the authors claim that higher dimensionality is better (l. 446)?
>
> - How exactly is the "prediction accuracy" of 73.02 reported in line 1880 (and in Fig. 21) calculated?
>
> - In Fig. 5, the max value of both D_eff and Psi_eff always match the same weight set (v1 or v2). Why are both used, then? I guess what I'm trying to understand is how do the results from Fig. 5 translate into the values provided in Fig. 21. For example ,Fig. 21a indicates that Psi_eff is better than D_eff on its own.
>
> - Fig. 21b shows that the pair Psi_eff + Within-class Mean Distance (WCD) actually outperforms the pair chosen by the authors. Is that because WCD gets right some of the cases where Psi_eff gets wrong? I'm stressing this point because, for people coming from a non-GLUE background, perhaps the most interesting thing about this paper is that it shows that many manifold-geometry metrics (GLUE or not) can be used to predict OOD performance, so it's important to precisely understand how this is being measured and what the potential differences between the various metrics might be. The results in Fig. 21 show that the simple correlation analysis from Fig. 4 is not a great predictor or their performance (e.g., participation ratio is highly correlated but has the lowest prediction accuracy).
>
> - Although you motivate and explain R_eff in the main text (an official GLUE metric), you never use it for prognostic prediction. Why not? Fig. 4 shows it correlates quite well. One would expect it should be used exactly like D_eff and Psi_eff (except in the opposite direction), yet its performance is not reported in Figure 21a nor b. Could the authors please share its prediction accuracy values?
>
> - I'm also curious about the two examples in Fig. 5 where the approach fails: ViT H_14 and ViT L_16 (due to acc. less than half of the 9 OOD datasets tested). Do others metrics have better success with those, or do they all fail?
>
> - Finally, there is no mention of Appendix D in the manuscript. Could you please add a reference to it in the main text?
>
> __2), 3), 4), 5)__
>
> Thank you for the clarifications.

---

> > ### Author Response · Authors · 2025-12-01
> > **Response to the response to rebuttal**
> >
> > We sincerely thank **reviewer YFoW26** for the exceptionally careful reading, constructive questions, and helpful suggestions. These comments significantly improved the clarity and rigor of both the main text and the supplementary materials. We have revised the manuscript accordingly, and all newly added or substantially modified passages in the main text and Appendix D are highlighted in **red**. Below, we address each point raised by the reviewer.
> >
> > **(1) Standard error estimation for D_eff and Psi_eff**
> >
> > We now clarify this procedure in Appendix D.3 and D.4.
> >
> > Specifically, for each marker we repeat the subsampling procedure (2 classes out of 1000; 50 examples out of ~100) **100 times**, compute the variance σ of the resulting 100 marker values, and report the standard error as σ / √100.
> >
> > **(2) Averaging of marker values in Fig. 5**
> >
> > Yes. Each marker value in Fig. 5 is the **average over 100 repetitions**. The standard errors shown are computed as described above. Details are now added to Appendix D.3 and D.4.
> >
> > **(3) Apparent negative correlation of D_eff in Fig. 4**
> >
> > We thank the reviewer for catching this.
> >
> > The quantity showing negative correlation is R_eff, not D_eff. In the previous version of Fig. 4, the column labeling could cause confusion. We have revised Fig. 4 for clarity and explicitly noted in the text which column corresponds to each marker.
> >
> > **(4) Definition of “prediction accuracy” (73.02%)**
> >
> > A detailed explanation is now included in Appendix D.3 and D.4.
> >
> > In short: using D_eff and Psi_eff jointly, a verdict was made for 126 cases (out of 180). Among these, 92 were correct, giving 73.02=92/126.
> >
> > **(5) Why use both D_eff and Psi_eff? Relationship between Fig. 5 and Fig. 21**
> >
> > Fig. 5 uses **both markers simultaneously**: we make a prediction only when *both* D_eff and Psi_eff agree on which weight set (v1 or v2) has the larger value.
> >
> > This criterion is satisfied for 14 out of 20 architectures (13 preferring v1; 1 preferring v2). Across the resulting 126 model–dataset cases, the joint rule correctly predicts 92 cases (73.02%).
> >
> > Fig. 21 explores the broader landscape of choices—including single-marker predictors (Fig. 21a) and two-marker combinations (Fig. 21b).
> >
> > Indeed, Psi_eff alone performs better than D_eff alone, but the combination D_eff + Psi_eff achieves even higher accuracy. We have clarified this in Appendix D.3 and D.4.
> >
> > **(6) Why does Psi_eff + WCD outperform the pair chosen by the authors in Fig. 21b?**
> >
> > We thank the reviewer for this insightful point.
> >
> > Different marker combinations succeed on different subsets of models, and in some cases WCD compensates for cases where Psi_eff alone errs. As noted, a key takeaway—especially for readers unfamiliar with GLUE—is that many manifold-geometry markers (GLUE or not) carry prognostic signal, even though simple correlation (Fig. 4) does not reliably predict their performance.
> >
> > For example, participation ratio shows high correlation yet low prediction accuracy.
> >
> > We now emphasize this point more clearly in Appendix D.3.
> >
> > **(7) Why was R_eff not included in Fig. 21?**
> >
> > As clarified in response to Question (3), R_eff is the 5th column from the right in Fig. 4 and shows poor correlation with OOD performance.
> >
> > We nonetheless evaluated its prognostic performance: used as a negative predictor, R_eff achieves **41.48%** accuracy (56 correct out of 135 applicable cases).
> >
> > We now report this in Appendix D.3 and have updated Fig. 4 to avoid confusion.
> >
> > **(8) Failure cases: ViT H_14 and ViT L_16 in Fig. 5**
> >
> > All markers—including GLUE and non-GLUE—fail to make correct predictions for these two architectures. This may indicate that large-ViT models possess representational geometries qualitatively different from the convolutional and smaller-transformer models in our study. We highlight this point as an interesting direction for future work.
> >
> > **(9) Missing reference to Appendix D**
> >
> > A pointer to Appendix D has now been added in the main text (lines 482–483).

---

### Official Review · Reviewer_H4Eg · 2025-10-31

**Soundness:** 3
**Presentation:** 3
**Contribution:** 2
**Rating:** 4
**Confidence:** 4

**Summary:**

The paper studies the question of predicting OOD performance of an image classification model having only the access to the in-distribution data. The authors draw parallels between the methodology applied in medicine to find meaningful biomarkers with significant predictive power about some phenomenon of interest. The parallel is exploited to find a set of analogous quantities for neural networks and their OOD performance based on the geometric structure of the neural network representations and the previously introduced GLUE framework. The paper argues for the approach using several experimental results suggesting the potential of the presented approach.

**Strengths:**

1. The topic is of great importance -- being able to reliably predict a model's OOD performance without having access to the OOD dataset is of great importance.
2. The presentation of the whole work is clear and interesting, with some room for improvement regarding the technical details (which I'll discuss later).
3. I particularly appreciate authors' debating current trends which overly focus on studying models using tools developed in mathematics or physics. The idea to draw more inspiration from other fields, e.g. medicine is an important one to keep a healthy level of exploration within the community. That being said, I don't feel that the analogy is particularly helpful in this case (I'll elaborate on this in the weaknesses section).
4. The claims made by the authors are mostly clearly supported by the provided and clearly presented experiments; I didn't have any feeling of the authors overclaiming their contributions.

**Weaknesses:**

1. The main issue I see with this work is the lack of comparison to previous works studying the problem. While the authors do reference several papers in related works, they seem to miss the core works that study the same questions. For instance, [1] introduced the Tunnel Effect Hypothesis, showing that the drop of OOD performance is strongly correlated with the numerical rank of representations. Further [2] refined the Tunnel Hypothesis, showing how the Tunnel Effect (and thus OOD performance) depends on multiple factors (e.g. image resolution, augmentation strength, architecture). While in this work, the authors study only 4 learning rates and 4 weight decays. Next [3, 4] study how one can influence the model's OOD performance by regularising for or against Neural Collapse, in particular [3] shows that OOD generalisation works against OOD detection. Finally [5] further confirms the findings from [1, 2, 3] and explains the mechanisms for increased or degraded OOD performance (something that the authors of this work purposefully set aside as they advocate for finding reliable markers first and trying to explain their mechanisms later).

 2. The work is heavily based on the GLUE framework, which is only briefly introduced in the main paper, and crucial aspects such as explanation of the effective dimension, effective radius,and  participation rate, are sent to the appendix -- these are the crucial objects
studied in this work and require a proper introduction (and justification) within the main text.

3. Limited scope of the experiments, while the experiments on image classfication are well executed I would expect more experiments focusing on different domains or different architectures especially given the fact that the experiments require training only linear classifier on top of representations, which can be easily collected from pretrained models which are freely available on repositories like HuggingFace or PyTorch.

4. The analogy with medicine "preclinical" studies is exaggerated. Testing the validity of the approach on smaller, simpler datasets and applying it on bigger, more complex datasets is what a typical workflow in ML experiments looks like, I don't feel we need to justify it with the medicine-based approach.

5. Overly used footnotes, please use them sparingly it makes it harder to follow the text when one has to jump to footnoote too often.


[1] The Tunnel Effect: Building Data Representations in Deep Neural Networks, https://arxiv.org/abs/2305.19753

[2] What Variables Affect Out-of-Distribution Generalization in Pretrained Models?, https://arxiv.org/abs/2405.15018

[3] Controlling Neural Collapse Enhances Out-of-Distribution Detection and Transfer Learning, https://arxiv.org/pdf/2502.10691

[4] NECO: NEURAL COLLAPSE BASED OUT-OF-DISTRIBUTION DETECTION, https://arxiv.org/pdf/2310.06823

[5] Unpacking Softmax: How Temperature Drives Representation Collapse, Compression, and Generalization, https://arxiv.org/abs/2506.01562

**Questions:**

1. How does the method compare to previously observed link between representations rank and ood performance?
2. What is the complexity of the method? How it depends on the number of samples used for the estimation of the quantities? How sensitive this method is to noise in both data and labels?
3. What are the weak points of the method? When can we expect this method to be a reliable source of information and when should we expect to break? How do these areas compare to using numerical rank?

---

> ### Author Response · Authors · 2025-11-20
> **Response part 1**
>
> We thank the reviewer for the detailed and constructive comments. We appreciate the recognition that predicting OOD performance *purely from in-distribution data* is an important and challenging question, and we are grateful for the thoughtful feedback on both conceptual and methodological aspects of our work. Your comments helped us broaden comparisons to prior work, clarify the presentation of GLUE-based measures, expand experimental coverage, and refine our framing. We have addressed each of these points carefully in the revision.
>
> Specifically, in response to this reviewer and others, we:
>
> (i) Toned down the **medical framing**, retaining only the conceptual motivation for diagnostic reasoning **(idQw, H4Eg, YFoW).**.
>
> (ii) Reduced the use of **footnotes**, moving content into the main text for readability (**H4Eg**).
>
> (iii) Added explicit comparisons to **tunnel Effect** and **neural collapse–based** indicators, including implementing these markers directly in our experiments (see Figure 4, line385-410 and **Line 162-185**)(**idQw, H4Eg**).
>
> (iv) Expanded the **technical introduction of GLUE markers** (effective dimension, utility, radius, and participation ratio) into the main text (**idQw, H4Eg**).
>
> (v) Substantially **expanded the pretrained-model experiments** to 20 architectures and 9 OOD datasets (Figure 5, line448-473), and evaluated on non-GLUE markers (**idQw, H4Eg)**.
>
> Below we respond point-by-point.
>
> ## **Detailed Response**
>
> ### **1) Lack of comparison to key prior works (Tunnel Effect, Neural collapse)**
>
> We appreciate the reviewer’s suggestion and have expanded the comparison in the revision (Lines 162–185).
>
> **Reviewer H4Eg** (the ref numbers from the original reviewer comment have been changed to the ref numbers used in this reply): “The main issue I see with this work is the lack of comparison to previous works studying the problem. […] For instance, [6] introduced the Tunnel Effect Hypothesis, showing that the drop of OOD performance is strongly correlated with the numerical rank of representations. Further [7] refined the Tunnel Hypothesis, showing how the Tunnel Effect (and thus OOD performance) depends on multiple factors (e.g. image resolution, augmentation strength, architecture). While in this work, the authors study only 4 learning rates and 4 weight decays. Next [8, 9] study how one can influence the model's OOD performance by regularising for or against Neural Collapse, in particular [8] shows that OOD generalisation works against OOD detection. Finally [5] further confirms the findings from [6, 7, 8] and explains the mechanisms for increased or degraded OOD performance (something that the authors of this work purposefully set aside as they advocate for finding reliable markers first and trying to explain their mechanisms later).”
>
> The key distinction is that both the Tunnel Effect and Neural Collapse lines of work study **how a representation changes across layers *within a single trained model***. Their central results demonstrate that OOD performance correlates with layerwise trends—e.g., decreasing numerical rank (Tunnel Effect [5,6,7]) or increasing neural collapse ([8,9]) as depth increases.
>
> Our setting is fundamentally different. We examine **differences across multiple models**, trained with different hyperparameters or initialization seeds but achieving similar ID accuracy. This cross-model comparison is significantly more challenging and is precisely the regime relevant for **model selection**, where practitioners must choose among many pretrained checkpoints without access to OOD data.
>
> To ensure fairness, we implemented the key metrics from these prior works—numerical rank (Tunnel Effect) and NC1/within-class distance (Neural Collapse)—and evaluated them in our **ID-only, cross-model prognostic setting** (Lines AA–BB). While these measures capture meaningful layerwise structure within individual models, they did **not** provide reliable predictions across different trained models, in contrast to the GLUE-based, task-relevant geometric markers that consistently performed well.
>
> This distinction is now explicitly clarified in the revised manuscript (line 162-185), along with quantitative comparisons for all metrics (Figure 4, line 385-410).

---

> > ### Author Response · Authors · 2025-11-20
> > **Response part 2**
> >
> > ### **2) Limited scope of experiments (domains and architectures)**
> >
> > We appreciate the reviewer highlighting this point.
> >
> > **Reviewer H4Eg**: “Limited scope of the experiments, while the experiments on image classfication are well executed I would expect more experiments focusing on different domains or different architectures especially given the fact that the experiments require training only linear classifier on top of representations, which can be easily collected from pretrained models which are freely available on repositories like HuggingFace or PyTorch.”
> >
> > In the revision, we substantially increased the scope by adding:
> >
> > - **20 pretrained ImageNet architectures** (including ResNet, WideResNet, RegNet, MobileNet, ViT, etc.)
> > - **9 class-level OOD datasets** (e.g., Places, Pets, Cars, Flowers, Aircraft, Food, Texture, Naturalist)
> >
> > This expanded evaluation tests variation in architecture, feature dimensionality, optimization dynamics, dataset scale, and pretraining strategy.
> >
> > In addition, we extended this comparison to the **ImageNet-pretrained model setting**, where we evaluated all markers on 20 architectures across 9 OOD datasets (Supplementary Figure 21, page 36). Although these markers performed less consistently than GLUE in the CIFAR experiments, several—such as participation ratio, numerical rank, and the Neural Collapse metric—still provided **substantial predictive power**, all significantly outperforming ID test accuracy as a prognostic indicator. This additional experiment demonstrates that even non-GLUE measures capture useful structure in complex, real-world pretrained models and further addresses the concern that our analysis relies solely on GLUE-based geometry.
> >
> > Regarding *other domains*, we agree that extending this diagnostic framework beyond image classification (e.g., language, multimodal, RL) is important.
> >
> > We now include a detailed discussion outlining how the methodology generalizes:
> >
> > 1. Identify a **failure mode** (e.g., hallucination in LLMs, grounding failures in multimodal models).
> > 2. Construct **matched models** with similar ID performance but different failure rates.
> > 3. Analyze their **internal representations** (token embeddings, sequence states, fusion-layer activations) to identify task-relevant markers.
> >
> > We treat this as a promising direction for future work rather than an empirical contribution of the present paper.
> >
> > ### **3) “Preclinical” medical analogy is exaggerated**
> >
> > We appreciate this feedback and agree that the analogy should remain conceptual.
> >
> > **Reviewer H4Eg**: “The analogy with medicine "preclinical" studies is exaggerated. Testing the validity of the approach on smaller, simpler datasets and applying it on bigger, more complex datasets is what a typical workflow in ML experiments looks like, I don't feel we need to justify it with the medicine-based approach.”
> >
> > In the revision, we have:
> >
> > - **Toned down the medical framing**,
> > - Removed uses of “preclinical,”
> > - And emphasized that our goal is to provide a **diagnostic perspective**, i.e., understanding *why* models fail, as opposed to pushing performance.
> > - Use the extra space to expand the explanation of the technical aspect (e.g., GLUE).
> >
> > The analogy is now lightweight and optional rather than central to the technical contribution.
> >
> > ### **4) Overuse of footnotes**
> >
> > Thank you for this suggestion.
> >
> > **Reviewer H4Eg**: “Overly used footnotes, please use them sparingly it makes it harder to follow the text when one has to jump to footnoote too often.”
> >
> > We reduced footnote usage substantially by integrating key explanations directly into the main text for clarity.

---

> > > ### Author Response · Authors · 2025-11-20
> > > **Response part 3**
> > >
> > > ### **5) Computational complexity and noise sensitivity**
> > >
> > > We thank the reviewer for asking.
> > >
> > > **Reviewer H4Eg**: “What is the complexity of the method? How it depends on the number of samples used for the estimation of the quantities? How sensitive this method is to noise in both data and labels?”
> > >
> > > The core computation in GLUE is solving quadratic programs with linear constraints. And the size of the problem is the number of features by the number of total points (i.e., the number of classes times the number of samples per class). As we downsampled the number of points per class to 50 in all our GLUE analyses and study pairwise manifold geometry, the size of the problem is #features x 100. And empirically, on a single CPU, one run of GLUE analysis can be done in 30 seconds.
> > >
> > > Regarding sensitivity to noise, GLUE is designed to average over random projections, which makes it **robust to small feature noise** (as shown in the original GLUE paper).
> > >
> > > ### **6) Weak points and when the method might break**
> > >
> > > We appreciate this question and address it transparently.
> > >
> > > **Reviewer H4Eg**: “What are the weak points of the method? When can we expect this method to be a reliable source of information and when should we expect to break? How do these areas compare to using numerical rank?”
> > >
> > > GLUE-based markers may be less informative when:
> > >
> > > - **Class structure is absent or heavily corrupted** (e.g., extreme label noise, self-supervised contrastive learning without pseudo-labels).
> > > - **The downstream task is not class separability–based**, e.g., dense prediction or multi-label tasks without clear manifolds.
> > > - **Models differ drastically in architecture**, where comparing absolute marker values is less meaningful.
> > >
> > > However, within **matched model families**—the intended use case—GLUE markers are consistently more stable and discriminative than raw rank or collapse measures.
> > >
> > > ## References
> > >
> > > [1] Distribution Shift Inversion for Out-of-Distribution Prediction: https://arxiv.org/abs/2306.08328
> > >
> > > [2] Predicting Out-of-Distribution Error with the Projection Norm: https://proceedings.mlr.press/v162/yu22i.html
> > >
> > > [3] To Annotate or Not? Predicting Performance Drop under Domain Shift: https://aclanthology.org/D19-1222/
> > >
> > > [4] Can you trust your model's uncertainty? Evaluating predictive uncertainty under dataset shift: https://arxiv.org/abs/1906.02530
> > >
> > > [5] Unpacking Softmax: How Temperature Drives Representation Collapse, Compression, and Generalization, https://arxiv.org/abs/2506.01562
> > >
> > > [6] The Tunnel Effect: Building Data Representations in Deep Neural Networks: https://arxiv.org/abs/2305.19753
> > >
> > > [7] What Variables Affect Out-of-Distribution Generalization in Pretrained Models?: https://arxiv.org/abs/2405.15018
> > >
> > > [8] Controlling Neural Collapse Enhances Out-of-Distribution Detection and Transfer Learning: [https://arxiv.org/pdf/2502.1091](https://arxiv.org/pdf/2502.10691)
> > >
> > > [9] NECO: NEURAL COLLAPSE BASED OUT-OF-DISTRIBUTION DETECTION, https://arxiv.org/pdf/2310.06823
> > >
> > > [10] Delays in generalization match delayed changes in representational geometry: https://proceedings.mlr.press/v285/zheng24a.html
> > >
> > > [11] Benchmarking Neural Network Robustness to Common Corruptions and Perturbations: https://arxiv.org/abs/1903.12261
> > >
> > > [12] Predicting With Confidence on Unseen Distributions: https://arxiv.org/abs/2107.03315

---

### Official Review · Reviewer_idQw · 2025-10-31

**Soundness:** 3
**Presentation:** 3
**Contribution:** 3
**Rating:** 4
**Confidence:** 4

**Summary:**

This paper introduces a top down, systems-level framework for understanding neural network failures inspired by similar approaches in biology and medicine. The framework is broken down into three stages: 1) identifying "biomarkers" or task-relevant measures of performance, 2) using biomarkers as prognostic tools in "clinical experiments" (i.e., to predict how markers identify failures to generalize), and 3) apply prognostic tools in a real-world application such as selecting the best pre-trained model for transfer learning. The "biomarkers" used in this paper lean heavily on prior work in representational geometry, specifically GLUE. They do a comprehensive empirical study to show that two measures from GLUE, the effective dimension and effective utility, reliably predict OOD performance better than less discriminative measures such as test accuracy. Finally, they apply their framework to the transfer learning setting they introduce earlier in the paper and show that geometric measures are more useful for predicting the best model to select.

**Strengths:**

1. The paper is extremely well-written, clear, and argues its central claims well. Despite relying heavily on prior work in GLUE, and therefore having little space to go over the theory, the authors do a good job of providing the necessary intuition for various concepts.
2. The empirical evaluation is comprehensive, covering many model architectures, datasets, and hyper-parameter configurations. Not only does this go a long way towards bolstering the authors claims, I believe the existence of this evaluation is useful to the community in and of itself.
3. Framing neural network interpretability work in the context of medical methodology is novel and potentially insightful.

**Weaknesses:**

1. While I appreciate the novelty of the medical framing, ultimately, none of the technical aspects of this framing translate into the framework. Instead of providing new insight, this perspective only seemed to confuse me. For example, the connection to "biomarkers" is far less important than emphasizing that measures of performance need to be task-relevant *as well as* descriptive of underlying mechanisms. The framing is not a deal-breaker, but I believe the paper would be stronger if it spent more time explaining what is unique about their measures w.r.t. predicting OOD generalization.
2. Though there is strong evidence in the paper for the GLUE-based framework under image classification, there is little evidence or discussion of how to extend the framework to other settings or measures. Overall, it feels as though this should have been a paper about the applications of GLUE for OOD.

**Questions:**

The paper's claim's can be broken down into:
1. A framework for neural network failure diagnosis based on top-down approaches.
2. A specific instance of (1) for image classification based on GLUE.

Q1. For Claim (1), little time is spent discussing how the framework should apply to settings other than image classification. Figure 1 is vague-enough that it could describe nearly any top down approach. And this is not the only top down approach for OOD performance prediction (see [1-5]). Can you explain what this framework proposes specifically to do in general?

Q2. For Claim (2), the paper focuses on GLUE, and it does a good job of explaining why the measures make sense for OOD prediction. But it isn't clear if this is the only measure that could work, or what properties of this measure make it effective. Based on the text, it seems that being task-relevant and discriminative are important, but then could [2] work? Are there alternatives?

Q3. Related to Q2, stronger evidence for Claim (1) would be to show something outside of GLUE that works and captures the essential properties of an effective biomarker.

Q4. The evaluation in Table 2 is limited to a set of models that where v1 and v2 differ in how much they train on the ID data. Could you test of a more diverse set of pre-trained models where it is less expected that the over-trained version performance worse OOD but better ID?

Q5: [6] seems to have noted the relationship between OOD performance and the measures in this paper; it is probably worth citing as related work.

[1] : https://openaccess.thecvf.com/content/CVPR2023/html/Yu_Distribution_Shift_Inversion_for_Out-of-Distribution_Prediction_CVPR_2023_paper.html

[2] : https://proceedings.mlr.press/v162/yu22i.html

[3] : https://openaccess.thecvf.com/content/ICCV2021/html/Guillory_Predicting_With_Confidence_on_Unseen_Distributions_ICCV_2021_paper.html

[4] : https://aclanthology.org/D19-1222/

[5] : https://proceedings.neurips.cc/paper/2019/hash/8558cb408c1d76621371888657d2eb1d-Abstract.html

[6] : https://openreview.net/pdf?id=1ae108kHk2

---

> ### Author Response · Authors · 2025-11-20
> **Response part 1**
>
> We thank the reviewer for the thoughtful and constructive comments. We appreciate the positive assessment of the paper’s clarity, empirical depth, and the overall value of studying geometric markers that anticipate OOD failures *before deployment*. The reviewer’s questions helped us clarify the conceptual framing of the diagnostic perspective, articulate more precisely how the framework generalizes beyond image classification, and highlight the rationale for using GLUE-based markers alongside several alternative baselines.
>
> In the revision, and in response to this reviewer and others, we made several substantial updates:
>
> (i) We **toned down the medical analogy** and expanded the technical framing, emphasizing the diagnostic mindset (understanding *why* failure happens) rather than performance-driven model design **(idQw, H4Eg, YFoW).**
>
> (ii) We added detailed discussions and comparisons with related work [1-5] **(idQw, H4Eg).**
>
> (iii) We added **additional non-GLUE markers** (AUROC [11], logit entropy [12], neural collapse metrics [8,9], tunnel effect metrics [6,7]), strengthening the comparison requested by the reviewer (Figure 4, line 385-410, also line 363-369, and line 162-185) **(idQw, H4Eg).**
>
> (iv) We expanded the set of **pretrained models and OOD datasets** to test model diversity more broadly (Figure 5, line 449-473). We further evaluated a broad set of non-GLUE markers—such as participation ratio, numerical rank, Neural Collapse measures, and logit-based scores—and found that they also achieved non-trivial prognostic performance, substantially outperforming ID test accuracy (Figure 21, Section D.4, line 1874-1904) **(idQw, H4Eg).**
>
> (v) We expanded the discussion on **framework generality**, including how to extend the diagnostic approach to language and multimodal models (Line 521-523) (**idQw, H4Eg, YFoW**).
>
> We hope the revisions address the reviewer’s concerns and help clarify the scope and significance of the diagnostic framework.
>
> ## **Detailed Response**
>
> ### **1) Medical framing and what is unique about our markers**
>
> We thank the reviewer for this insightful point.
>
> **Reviewer idQw:** “While I appreciate the novelty of the medical framing, ultimately, none of the technical aspects of this framing translate into the framework. Instead of providing new insight, this perspective only seemed to confuse me. For example, the connection to "biomarkers" is far less important than emphasizing that measures of performance need to be task-relevant *as well as* descriptive of underlying mechanisms.”
>
> We agree that the analogy is conceptual and not technically necessary, and we have accordingly **reduced the emphasis on medical terminology**. We now use it sparingly only as intuition-building language.
>
> The key idea we aim to emphasize—now made explicit in the revision—is that the **diagnostic perspective is structurally different from the dominant ML paradigm**. Standard machine-learning practice aims to design models that *perform better*. In contrast, our goal is to **understand when and why a model fails**, by studying learned representations and identifying task-relevant markers associated with specific failure modes.
>
> We expanded the exposition to highlight the unique properties of our markers:
>
> - They are **task-relevant**, grounded in the geometric structure of classification manifolds.
> - They detect **over-specialization** on ID data, which is not visible through accuracy alone.
> - They are **ID-only** predictors, unlike OOD-detection methods that typically require access to shifted samples.
> - They generalize across diverse architectures and training regimes.
>
> We also reworded several passages (Line 162-185) to make these distinctions clearer.

---

> > ### Author Response · Authors · 2025-11-20
> > **Response part 2**
> >
> > ### **2) Clarifying how our diagnostic analysis on OOD generalization differs from previous methods**
> >
> > **Reviewer idQw** (the ref numbers from the original reviewer comment have been changed to the ref numbers used in this reply)**:** “this is not the only top down approach for OOD performance prediction (see [1,2,12,3,4]). Can you explain what this framework proposes specifically to do in general?”
> >
> > In the revision, we clarified how our diagnostic framework differs from prior OOD-prediction approaches that the reviewer mentioned. Traditional OOD detection methods aim to distinguish OOD samples from ID samples by analyzing differences in logits or feature statistics (Line 162-185). These methods typically address “**label-preserving distribution shifts”** [1,2,12,3,4], where the input distribution changes but the class labels remain the same. By contrast, “**class-level OOD generalization”**—the setting we study—is substantially more challenging because the target task contains entirely unseen classes. Performance in this setting is commonly evaluated by training a linear probe on penultimate-layer features.
> >
> > Several recent works analyze representational factors relevant to this probe-based OOD performance. The Tunnel Effect papers [6,7] relate OOD accuracy drops **across layers** to decreases in the **numerical rank** of OOD features, while Neural Collapse work [8,9] examines how collapse measures (e.g., NC1) vary **across layers within a single model**. As we now emphasize in the manuscript (Lines 175-185), these approaches investigate *layerwise representational evolution inside one network*, not variation **across different models or training configurations**.
> >
> > Our diagnostic analysis focuses precisely on this cross-model regime. We study how representational geometry differs among models trained with different hyperparameters or initialization seeds, holding ID accuracy approximately constant. This setting is the one relevant for **model selection** and **prognostic prediction**, and it is not addressed by prior layerwise analyses. In the revision (Lines 162–185), we also added text distinguishing our framework from existing top-down OOD-prediction approaches ([1-9]).
> >
> > Finally, we incorporated ideas from these prior works into our marker suite: numerical rank (Tunnel Effect), feature-collapse metrics (Neural Collapse), and several logits-based scores (Figure 4), see details below.

---

> > > ### Author Response · Authors · 2025-11-20
> > > **Response part 3**
> > >
> > > ### **3) GLUE vs alternatives; are there other effective markers?**
> > >
> > > We thank the reviewer for raising this important conceptual question.
> > >
> > > **Reviewer idQw** (the ref numbers from the original reviewer comment have been changed to the ref numbers used in this reply)**:** “it isn't clear if this is the only measure that could work, or what properties of this measure make it effective. Based on the text, it seems that being task-relevant and discriminative are important, but then could [2] work? Are there alternatives?”
> > >
> > > The reviewer noted that task-relevant and discriminative measures seem promising and asked whether previous OOD prediction methods could be used. It is important to clarify that methods in this family, including [2], **require access to OOD samples** (or their statistics) in order to estimate shifted distributions or construct predictive proxies. In contrast, our diagnostic setting is deliberately **strictly harder**: the predictor must be computed **using ID data only**, with no knowledge of what the future OOD task will be. This difference is fundamental—methods that rely on OOD examples or signals cannot be directly applied in our setting.
> > >
> > > Historically, GLUE was chosen because prior work on representational geometry suggested a close connection between **manifold separability**, **task-relevant variability**, and **generalization behavior**. However, we did not assume GLUE would work best. Before committing to it, we systematically evaluated a broad range of alternative markers—now expanded in the revision:
> > >
> > > - Participation ratio (PR)
> > > - Covariance-based statistics
> > > - Activation sparsity
> > > - Pairwise distance/angle distributions
> > > - (new in revision) Logit-based OOD markers (AUROC [11], logit entropy [12])
> > > - (new in revision) Neural collapse metric [8,9]
> > > - (new in revision) Tunnel effect metric (numerical rank) [6,7]
> > >
> > > We now explicitly report these baselines in the revised Section 3 and present the results in Figure 4.
> > >
> > > In addition, we extended this comparison to the **ImageNet-pretrained model setting**, where we evaluated all markers on 20 architectures across 9 OOD datasets (Supplementary Figure 21, page 36). Although these markers performed less consistently than GLUE in the CIFAR experiments, several—such as participation ratio, numerical rank, and the Neural Collapse metric—still provided **substantial predictive power**, all significantly outperforming ID test accuracy as a prognostic indicator. This additional experiment demonstrates that even non-GLUE measures capture useful structure in complex, real-world pretrained models and further addresses the concern that our analysis relies solely on GLUE-based geometry.
> > >
> > > We found:
> > >
> > > - **PR and numerical rank often perform reasonably well**, but is not consistent across all conditions.
> > > - **Logit-based markers perform poorly** in the class-level OOD setting because they implicitly assume access to shifted data.
> > > - **Neural collapse measures correlate in some cases**, but again are not consistent across all architectures/OOD tasks.
> > >
> > > In contrast, D_eff and Psi_eff were the only markers that remained predictive *across architectures, datasets, pretraining regimes, and OOD targets*.
> > >
> > > We emphasize in the rebuttal that GLUE is **not the only possible option**, but rather a principled task-relevant instantiation for classification manifolds. Other domains will require their own task-specific markers.
> > >
> > > ### **4) Diversity of pretrained models**
> > >
> > > We thank the reviewer for this helpful suggestion.
> > >
> > > **Reviewer idQw:** “The evaluation in Table 2 is limited to a set of models that where v1 and v2 differ in how much they train on the ID data. Could you test of a more diverse set of pre-trained models where it is less expected that the over-trained version performance worse OOD but better ID?”
> > >
> > > In the revision, we expanded our pretrained model evaluation to include **20 architectures** and **9 OOD target datasets** (Figure 5). Each model has two released versions (v1 and v2). By design, the v2 weights achieve higher accuracy on the in-distribution ImageNet benchmark. Our prognostic indicators predicted that v1 would outperform v2 on OOD transfer in 11 cases (despite v2 having higher ID accuracy), that v2 would outperform v1 in 2 cases, and yielded no clear verdict for the remainder. See Figure 5 for results. Among these 15 models and 9 OOD datasets, our prediction accuracy is 73.02% (92 out of 126). This is much higher than using ID test accuracy as a predictor for OOD performance (37.22%).  Also, using some of the other markers (e.g., Neural Collapse, Participation Ratio) also yields non-trivial prediction accuracy in OOD performance than using ID test accuracy (Figure 21, line 1890-1904).
> > >
> > > We also clarified that the comparison made here is “between different pretrained models”, while the neural collapse [8,9] and tunnel effect [6,7] work focuses on correlations between OOD performance and their measures across the layers of a single model (line 175-185).

---

> > > > ### Author Response · Authors · 2025-11-20
> > > > **Response part 4**
> > > >
> > > > ### **5) Extending the framework to other settings**
> > > >
> > > > We agree with the reviewer that the generality of the framework deserves further clarification.
> > > >
> > > > **Reviewer idQw:** “little time is spent discussing how the framework should apply to settings other than image classification.”
> > > >
> > > > While the current paper instantiates the framework in image classification using GLUE, the **methodological principles are task-agnostic**:
> > > >
> > > > 1. **Identify a failure mode** of interest in the target domain (e.g., hallucination in LLMs, grounding errors in multimodal models).
> > > > 2. **Construct matched model pairs or families** that have similar ID performance but differ in failure rate.
> > > > 3. **Analyze their internal representations** (e.g., token embeddings, sequence representations, fusion-layer embeddings) to search for markers correlated with the failure mode.
> > > > 4. **Validate these markers** across datasets, architectures, or training recipes.
> > > >
> > > > We added a dedicated paragraph describing how this process applies to **language** and **multimodal** models (Line 521-523 in the revision). This complements the reviewer’s suggestion and clarifies Claim (1) independently of GLUE.
> > > >
> > > > ## References
> > > >
> > > > [1] Distribution Shift Inversion for Out-of-Distribution Prediction: https://arxiv.org/abs/2306.08328
> > > >
> > > > [2] Predicting Out-of-Distribution Error with the Projection Norm: https://proceedings.mlr.press/v162/yu22i.html
> > > >
> > > > [3] To Annotate or Not? Predicting Performance Drop under Domain Shift: https://aclanthology.org/D19-1222/
> > > >
> > > > [4] Can you trust your model's uncertainty? Evaluating predictive uncertainty under dataset shift: https://arxiv.org/abs/1906.02530
> > > >
> > > > [5] Unpacking Softmax: How Temperature Drives Representation Collapse, Compression, and Generalization, https://arxiv.org/abs/2506.01562
> > > >
> > > > [6] The Tunnel Effect: Building Data Representations in Deep Neural Networks: https://arxiv.org/abs/2305.19753
> > > >
> > > > [7] What Variables Affect Out-of-Distribution Generalization in Pretrained Models?: https://arxiv.org/abs/2405.15018
> > > >
> > > > [8] Controlling Neural Collapse Enhances Out-of-Distribution Detection and Transfer Learning: [https://arxiv.org/pdf/2502.1091](https://arxiv.org/pdf/2502.10691)
> > > >
> > > > [9] NECO: NEURAL COLLAPSE BASED OUT-OF-DISTRIBUTION DETECTION, https://arxiv.org/pdf/2310.06823
> > > >
> > > > [10] Delays in generalization match delayed changes in representational geometry: https://proceedings.mlr.press/v285/zheng24a.html
> > > >
> > > > [11] Benchmarking Neural Network Robustness to Common Corruptions and Perturbations: https://arxiv.org/abs/1903.12261
> > > >
> > > > [12] Predicting With Confidence on Unseen Distributions: https://arxiv.org/abs/2107.03315

---

> > > > > ### Comment · Reviewer_idQw · 2025-11-24
> > > > > **Response to rebuttal**
> > > > >
> > > > > Thank you for the detailed response and changes to the paper. Overall, your response and corresponding changes have sharpened my understanding of the goals/claims of the work (particularly the ID vs OOD distinction). I believe future readers will also benefit from the text changes. I have some concerns that are listed below, but with the current changes, I believe the paper is stronger and the evidence supports its central arguments better.
> > > > >
> > > > > 1. The authors write (lines 362-364): "alternative markers—including Neural Collapse, numerical rank, and logics-based OOD-detection scores—did not perform as reliably". This is not what I read from Fig. 4. While logic-based measures are indeed worse, neural collapse, within-class distances, and numerical rank all seem to perform reasonably well. In particular, numerical rank is on par or better than effective utility.
> > > > > 2. Relatedly, Fig. 4 and Fig. 21 both make it so that the results move beyond GLUE. There are other markers that also work within the framework. So, what is the contribution of this paper? A conceptual framework for diagnosing when and why a model might fail, or an empirical study of GLUE measures and their correlation with OOD generalization? Currently, these two claims are muddled, and I think your introduction and evidence supports the first more clearly.
> > > > > 3. You might consider editing Fig 1. (middle) to highlight that prognostic discovery uses ID data to make OOD predictions (complementing the changes to the text).
> > > > > 4. Typo: line 062 uses ID / OOD before they are introduced
> > > > >
> > > > > Overall, I think this is strong empirical work that can be useful to the field, but as written it is difficult to endorse since the evidence and claims are mismatched. I think the authors are over-emphasizing GLUE in the text (sometimes incorrectly, see (1) above). More confusingly, the central contribution (Fig. 1) in the abstract and introduction don't need GLUE to be supported and count as a contribution. GLUE can still be the focused markers in the study, and the text adjusted to de-emphasize GLUE and acknowledge other markers. In doing so, I think the authors would support their main contribution (Fig. 1) more strongly.

---

> > > > > > ### Author Response · Authors · 2025-11-25
> > > > > > **Response to the response to rebuttal (part 1/2)**
> > > > > >
> > > > > > We thank reviewer **idQw** for the thoughtful follow-up comments and for engaging deeply with our revision. We are very glad to hear that the updated text clarified our goals—particularly the ID-only vs. OOD prediction distinction—and we appreciate the additional suggestions, all of which help strengthen the alignment between claims and evidence. We also fully agree with the reviewer that the methodological framework and our empirical findings are the central contribution of the paper and deserves to be emphasized more prominently than specific markers such as GLUE; we will revise the text accordingly to make this priority clearer.
> > > > > >
> > > > > > Below we respond to each point in turn, and then summarize the edits we plan to make in the next revision.
> > > > > >
> > > > > > ### **(1) Clarifying the performance of numerical rank and Neural Collapse in Fig. 4**
> > > > > >
> > > > > > We thank reviewer **idQw** for this correction. Our previous wording (“did not perform as reliably”) was indeed too strong.
> > > > > >
> > > > > > We agree that numerical rank and Neural Collapse (NC1 / within-class distance) perform reasonably well, sometimes comparable to effective utility, especially when evaluated within the same model family. What distinguishes D_eff and Psi_eff in Fig. 4 is that they remain predictive across heterogeneous architectures and training regimes, whereas numerical rank exhibits weaker statistical strength in a few cases (e.g., VGG-19 SGD).
> > > > > >
> > > > > > We will revise the text to be more accurate and balanced. In particular, we will emphasize that:
> > > > > >
> > > > > > - Numerical rank does often work well except few cases (VGG-19, VGG-16),
> > > > > > - GLUE’s measures exhibit slightly greater consistency across settings,
> > > > > > - and many non-GLUE markers also show valuable prognostic power (Fig. 4, Fig. 21).
> > > > > >
> > > > > > We will adjust the interpretations of the results in Fig. 4 (Lines 359–374) in the next revision, see the end of this response thread for details.
> > > > > >
> > > > > > ### **(2) Clarifying the paper’s contribution: framework vs. GLUE**
> > > > > >
> > > > > > We thank reviewer **idQw** for this very insightful clarification. Our *original intention* was indeed to highlight the diagnostic framework (Fig. 1) as the central contribution, with GLUE serving only as one instantiation of the framework for image classification.
> > > > > >
> > > > > > We appreciate the reviewer pointing out that the current draft unintentionally over-emphasizes GLUE, which may blur the methodological contribution. We are happy to de-emphasize GLUE and make the intended contribution clearer.
> > > > > >
> > > > > > In the next revision, we will (see the end of this response thread for details):
> > > > > >
> > > > > > - Rework the introduction (Lines 91–103) to foreground the diagnostic methodology, not GLUE;
> > > > > > - Clarify that the main contribution is the general top-down diagnostic framework for identifying failure modes using ID-only information;
> > > > > > - Explicitly recognize that other markers—numerical rank, Neural Collapse, within-class distance—are also valuable under this framework (Fig. 4, Fig. 21);
> > > > > > - Frame GLUE as a strong but non-exclusive example of a task-relevant geometric marker, not as the conceptual centerpiece.
> > > > > >
> > > > > > We believe these changes will make the paper’s contribution more coherent and fully aligned with our original goals.
> > > > > >
> > > > > > ### **(3) Editing Fig. 1 (middle panel)**
> > > > > >
> > > > > > We agree with the suggestion.
> > > > > >
> > > > > > We will revise the “Prognostic Discovery” panel of Fig. 1 to explicitly highlight that the prediction is made using ID data only and the OOD task is unknown at diagnostic time—mirroring the clarified text in the introduction.
> > > > > >
> > > > > > ### **(4) Typo at line 062**
> > > > > >
> > > > > > We will fix this typographical error and ensure that ID/OOD terminology is introduced only after definitions appear.

---

> > > > > > > ### Author Response · Authors · 2025-11-25
> > > > > > > **Response to the response to rebuttal (part 2/2)**
> > > > > > >
> > > > > > > # **Planned Revisions (will update the manuscript after other reviewers’ replies)**
> > > > > > >
> > > > > > > Although we will wait for all remaining reviewer responses before uploading a new PDF (so that line numbers in previous replies remain valid), here is a summary of edits we plan to implement:
> > > > > > >
> > > > > > > - **Figure 1 (middle panel):** highlight the “ID-only prediction” aspect.
> > > > > > >     - “Identify prognostic patterns of markers linked to failure modes in medium-scale experiments” → “Identify prognostic patterns of **ID** markers linked to **OOD** failure modes in medium-scale experiments”
> > > > > > > - **Lines 93–103:** modify the overview-of-contributions section to emphasize the framework, de-emphasize GLUE.
> > > > > > >     - “For image classification, we adopt the Geometry Linked to Untangling Efficiency (GLUE) […] providing useful intuitions for interpreting their values and results” → “A central step in our diagnostic framework is selecting and designing markers—scalar quantities computed entirely from ID data—that capture aspects of a pretrained model relevant for downstream generalization. In image classification, we focus on penultimate-layer feature vectors, as the final classification decision is obtained by a linear readout from this layer. Accordingly, we evaluate a broad family of candidate markers, including: (i) accuracy- and logits-based quantities, (ii) low-order statistical summaries of representations (e.g., sparsity, covariance structure), and (iii) geometric measures of class-conditioned point-cloud manifolds—such as participation ratio, within-class spread, neural-collapse metrics (Papyan et al., 2020; Harun et al., 2025), numerical rank (Masarczyk et al., 2023; Harun et al., 2024), and task-relevant geometric measures from the GLUE framework (Chou et al., 2025a).”
> > > > > > > - **Section 2 (Markers for Image Classification)**: We will swap the order of subsections, first introducing the conventional measures, and then explain the task-relevant geometric measures.
> > > > > > > - **Lines 359–374:** adjust interpretation of Fig. 4 to more fairly acknowledge strong performance from numerical rank.
> > > > > > >     - “whereas alternative markers—including Neural Collapse (Harun et al., 2025), numerical rank (Tunnel Effect (Masarczyk et al., 2023)), and logits-based OOD-detection scores—did not perform as reliably.” → “[…] OOD-detection scores—showed statistically weaker or less consistently predictive trends across settings, with numerical rank performing well in most but a few cases (e.g., VGG-19 using SGD).”
> > > > > > > - **Line 062:** fix a premature use of ID/OOD.
> > > > > > > - **Line 487-488**: add concrete numbers on the results of using other markers (e.g., Neural Collapse, Participation Ratio) in the ImageNet experiment.
> > > > > > >     - “We remark that using some of the other markers (e.g., Neural Collapse, Participation Ratio) also yields non-trivial prediction accuracy in OOD performance.” → “We remark that using some of the other markers (e.g., Neural Collapse, Participation Ratio) also yields non-trivial prediction accuracy in OOD performance (e.g., using effective utility + within-class mean distance can achieve 76.77% prediction accuracy).”
> > > > > > >
> > > > > > > We thank reviewer **idQw** for the helpful feedback, and we warmly welcome any additional suggestions for improving clarity or strengthening the manuscript.

---

> > > > > > > > ### Comment · Reviewer_idQw · 2025-11-25
> > > > > > > >
> > > > > > > > Thank you for the detailed changes to my suggestions. I believe the paper is stronger and will increase my score to reflect that. My reasons for increase are:
> > > > > > > > - Text changes that clarify the ID-only data distinction, which I believe makes the goals of this work unique in comparison to prior approaches
> > > > > > > > - Text changes to center and clarify the main contribution as a methodological framework; acknowledgement of non-GLUE markers that can work within this framework
> > > > > > > > - Expanded evaluation to include other markers that fit within the framework
> > > > > > > > - Expanded evaluation for the real world example to include other architectures where the rule "overfit on train implies poor generalization" is not always the case, yet the proposed markers predict this
> > > > > > > > - Discussion of how the framework can be applied to other settings not studied in the paper

---

### Official Review · Reviewer_qBkT · 2025-10-31

**Soundness:** 3
**Presentation:** 4
**Contribution:** 4
**Rating:** 10
**Confidence:** 4

**Summary:**

The manuscript attempts to identify markers — statistics of the learned representations — that would help predict OOD failures. Comparing multiple marker candidates across multiple datasets, it concludes that object manifold dimension and utility are prime candidates for this. In a follow-up experiment, it is demonstrated that indeed they serve this purpose well.

**Strengths:**

* Fantastic scientific exposition of the idea, the design, and the results.
* The found candidates for OOD failure markers are interesting and non-trivial. Thus, it may trigger future research on the mechanism beyond their contribution to identifying OOD failures, leading to a breakthrough in our understanding of the issue.

**Weaknesses:**

* The manuscript is performing what is called in statistical literature "a fishing expedition" for markers. For a fixed set of datasets, had the author tested thousands of markers, they could have reported selectively on the most promising ones, thus finding markers that succeed by chance and do not generalize to other datasets. I don't suspect the authors' ethics -- but they need to take measures against such a mistake. The authors can safeguard against random markers by calculating the number of markers for which the achieved result would have happened by change, showing they are well below it (e.g. "this would have happened by change in probability 1/200, and we only tested 20 markers").
 * The main result, that a reduced manifold dimension is associated with poor generalization, is somewhat paradoxical. Previous literature cited in "related work" reports a reduction in manifold dimension across layers of a deep network, which is associated with improved classification performance. Indeed, if an object were to collapse to a point, ID performance would peak, and OOD performance would diminish. On the other hand, an uninitialized deep network where the manifold dimension is very high would perform equally bad in ID and OOD tasks. The authors should acknowledge and address this weakness openly.
 * The authors' explanation for why these specific markers are best (section 3.3) seems post-hoc and premature.

**Questions:**

* Can you suggest what the optimal levels of the candidate markers are? Or an educated procedure on how to operationalize their use in practice?

---

> ### Author Response · Authors · 2025-11-20
> **Response part 1**
>
> We thank the reviewer for the thoughtful, encouraging, and highly constructive feedback. We appreciate the positive assessment of our exposition, our scientific motivation, and the novelty of identifying task-relevant geometric markers for predicting OOD failures. The reviewer’s comments and questions helped us clarify several underlying assumptions, articulate the scope of our claims more carefully, and strengthen both the experimental evaluation and the conceptual framing of the paper.
>
> In the revised manuscript, we have incorporated a substantial set of improvements motivated by the reviewer’s insights as well as related points raised by the other reviewers:
>
> **(i) Expanded explanation of the diagnostic framework (idQw, H4Eg, YFoW).**
>
> We added several paragraphs that clarify (a) what distinguishes our setting from prior OOD-generalization work (line 162-185), and (b) how the framework extends beyond image classification (line 521-523).
>
> **(ii) Comparison with existing OOD-detection methods (idQw, H4Eg).**
>
> Following reviewer suggestions, we incorporated several conventional OOD-style markers (logit entropy, AUROC-style scores) and evaluated them alongside our geometric metrics. This helps position GLUE-based markers relative to widely used baselines (Figure 4, line 385-410).
>
> **(iii) Discussion of Neural Collapse and the Tunnel Effect (H4Eg).**
>
> We explicitly discuss the connection and differences between GLUE’s geometric signals and the neural collapse / tunneling-effect literature (line 162-185), and we implemented within-class collapse scores as additional markers in our experiments (Section 3, Section D4).
>
> **(iv) Expanded analysis on pretrained models (idQw, H4Eg).**
>
> We clarified why the v1–v2 model pairs represent a meaningful and challenging benchmark and added further explanation about the diversity of architectures considered (Figure 5, line 449-473).
>
> We hope these revisions address the reviewer’s remaining concerns while further strengthening the contributions and clarity of the paper.
>
> ## **Detailed Response**
>
> ### **1) “Fishing expedition” comment**
>
> We appreciate the reviewer raising this important statistical point:
>
> **Review qBkT:** “For a fixed set of datasets, had the author tested thousands of markers, they could have reported selectively on the most promising ones, thus finding markers that succeed by chance and do not generalize to other datasets.”
>
> Our study evaluated **12 primary markers**, each with a small number of natural variants (e.g., global vs. per-class evaluation), and in the revision, we added **four additional markers** inspired by the reviewers’ suggestions (logit-based OOD scores, neural collapse metrics [1,2], and tunnel effect metrics [3]). All selected markers belong to well-established families used in prior literature on representation analysis (e.g., PR, covariance), OOD detection (e.g., AUROC [4], logit entropy [5]), or prior theoretical work motivating GLUE’s geometry.
>
> We did **not** perform a broad or unconstrained search over thousands of candidate functions. Instead, our marker set is:
>
> - **principled** (rooted in existing representational and OOD literature),
> - **task-motivated** (all markers reflect known structural attributes of feature geometry or logits), and
> - **limited in scope** (≈ 16 total after revision).
>
> Furthermore, we explicitly report **failures** of several markers, including participation ratio, covariance statistics, and logits-based measures. In fact, PR and Neural Collapse metric perform reasonably well in many cases—consistent with prior neuroscience results that PR is sometimes, but not always, a reliable proxy. This strengthens the argument that the GLUE-derived D_eff and Psi_eff provide a **more robust and task-relevant** measure across diverse settings.

---

> > ### Author Response · Authors · 2025-11-20
> > **Response part 2**
> >
> > ### **2) “Paradox” of manifold dimension and generalization**
> >
> > The reviewer noted that “an uninitialized deep network where the manifold dimension is very high would perform equally bad in ID and OOD tasks.” We agree, and we appreciate the opportunity to clarify this point.
> >
> > It is indeed true that
> >
> > - an *uninitialized* or poorly trained model exhibits **high manifold dimension** and performs poorly on both ID and OOD tasks—a phenomenon also observed in recent analyses of feature learning dynamics (e.g., lazy learning, see also Figure 7 in [6]).
> > - conversely, very deep or highly specialized networks may undergo **excessive compression**, achieving strong ID accuracy while displaying degraded OOD generalization.
> >
> > Our findings should therefore be interpreted **conditional on successful feature learning**, i.e., when comparing models with comparable ID performance. Within this ID-matched regime, a **larger effective dimension corresponds to richer, task-relevant variability** in the learned manifolds and predicts better transferability, whereas excessive compression reflects over-specialization to the ID distribution.
> >
> > In the revised manuscript (Lines 430–433), we clarify this distinction between the *absolute* dimensionality of untrained representations and the *effective, task-relevant* dimensionality of learned manifolds, emphasizing that our conclusions apply specifically within the “ID-matched, learned-feature” setting.
> >
> > ### **3) Post-hoc nature of the explanation in Section 3.3**
> >
> > We thank the reviewer for this important point.
> >
> > **Review qBkT:** “The authors' explanation for why these specific markers are best (section 3.3) seems post-hoc and premature.”
> >
> > We agree that the mechanistic explanation in Section 3.3 is **hypothesis-generating**, not causal. The findings are primarily **empirical**, motivated by the GLUE theory’s formal connection between manifold geometry and linear separability.
> >
> > ### **4) Optimal levels of markers / practical usage**
> >
> > We appreciate the reviewer’s insightful question.
> >
> > **Review qBkT:** “Can you suggest what the optimal levels of the candidate markers are? Or an educated procedure on how to operationalize their use in practice?”
> >
> > Because tasks differ in structure and complexity (classification, hierarchical labeling, reinforcement learning, and regression), absolute thresholds are unlikely to be universally applicable. Instead, we provide practical guidance compatible with our findings:
> >
> > - For **model selection within a class of pretrained networks**, comparing **relative** values of D_eff and Psi_eff is effective: *higher* values typically signal better OOD performance among models with similar ID accuracy.
> > - For **new domains or tasks**, marker structure should reflect task structure: e.g., classification tasks benefit from class-wise geometry; continuous tasks may require GLUE-inspired measures adapted to regression manifolds.
> > - A practical diagnostic workflow is included in the revision: compute penultimate features, run the GLUE metrics, compare marker values across candidate models, then optionally combine with conventional statistics (PR, collapse scores, logits-based metrics).
> >
> > We emphasize that GLUE provides a **task-relevant, mechanistically grounded starting point**, and we expect markers for other task families to arise from analogous theoretical analyses.
> >
> > ## References:
> >
> > [1] Controlling Neural Collapse Enhances Out-of-Distribution Detection and Transfer Learning: https://arxiv.org/pdf/2502.10691
> >
> > [2] What Variables Affect Out-of-Distribution Generalization in Pretrained Models?: https://arxiv.org/abs/2405.15018
> >
> > [3] The Tunnel Effect: Building Data Representations in Deep Neural Networks: https://arxiv.org/abs/2305.19753
> >
> > [4] Benchmarking Neural Network Robustness to Common Corruptions and Perturbations: https://arxiv.org/abs/1903.12261
> >
> > [5] Predicting with Confidence on Unseen Distributions: https://arxiv.org/abs/2107.03315
> >
> > [6] Feature Learning beyond the Lazy-Rich Dichotomy: Insights from Representational Geometry: https://arxiv.org/abs/2503.18114

---

> > > ### Comment · Reviewer_qBkT · 2025-11-25
> > > **Reponse to authors**
> > >
> > > I appreciate your detailed answers and improvement of the manuscript and stand by my initial (high) rating.

---

### Meta-Review · Area_Chair_QMcj · 2026-01-07

**Summary:**

The goal of this work is to identify when generalization errors occur in deep learning systems by analyzing the internal representation structure (geometry). The authors test a number of metrics, based in prior theory of what a "good" representation should be, and find that there are a few highly informative features that are highly predictive of out-of-distribution generalization (or lack thereof). The authors tested their hypothesis on a number of datasets and benchmarked against many of the relevant prior art.

The reviewers were at first mixed, with concerns ranging from missing benchmarks, confusion over the framing, and perceived limitations of the experiments. The authors added significant content and extensive edits that I feel addressed most of them, however I'm not entirely sure that one of the more lukewarm reviewers would have been convinced. I would have liked to know if the additional clarification and metrics included meth the expectations of reviewer H4Eg, since I did find that this was a nuanced and tricky (and important) point that would be good to have fully ironed out.

Given that the other reviewers were slightly (and in the case of one reviewer very) positive, I'm recommending acceptance.

**Reviewer Concerns:**

1) Paradox between low manifold dimension being a positive (in past work) vs. negative (in this work) indicator of good generalization
2) The number of features tested and if this constitutes a "fishing expedition" that needs be mitigated against finding a feature by chance
3) limitation of the approach to be overly focused on image classification
4) limitations of the experiments to only obviously overtrained models
5) Clarity, including confusion around the "medical diagnosis" framing of the method
6) Lack of comparison to other methods
7) Over-reliance on the GLUE framework (as well as lack of clarity on the full use of GLUE)

To address these concerns the authors heavily edited the text to provide explanations of the method and tone down the emphasis on the medical diagnosis framing. This, and the discussion with reviewers addressed both the paradox (1), the issue with "fishing for features" (2), clarity of the task-agnostic nature of the model (4), and clarity of the framing (5). The authors also significantly extended their experiments to include many more pre-trained models, addressing (4). To address (6), the authors further clarified the difference in problem setting between their work and prior work that was mentioned by the reviewer. Moreover, the authors implemented metrics based on those prior models and showed reduced performance in predicting out-of-distribution generalization. Finally, the authors tested additional markers to compare against GLUE and show the utility of GLUE.

These efforts by the authors I feel addressed most of the original concerns. There were additional concerns raised by reviewers, in particular reviewer YFoW in the discussion as they understood the methods and scope better. In particular related to details in the paper's results, such as missing details about how different quantities were computed, errors in plot labels, and a few others. The authors did have a chance to post a reply, which seemed to clarify most of these points.

**Reviewer Scores:**

Initially this paper received scores of 4,10,6,4. Reviewer qBkT stated that they intended to retain their score of "10". Reviewer idQw indicated that they would raise their score, which I assume is to a 6 (they did not specify to what).

Reviewer YFoW had initially seemed to have a lot of confusion about the framing, which led to what seems like a shorter initial review that had a numerical score of 6. This reviewer was responsive, however, after the authors replied and had asked a number of clarifying questions. Overall this reviewer seemed a bit skeptical, although positive, and so my reading is that they would have stayed at 6.

Reviewer H4Eg was the only one that did not respond in time. I would have liked to see if the comparisons to prior work that were added met their expectations. This is teh biggest uncertainty factor for this paper. I think there's maybe a 50/50 chance they would have increased to a 6.

---

### Decision · Program_Chairs · 2026-01-26

Accept (Poster)